# Increased generalisation in trait anxiety is driven by aversive value transfer
Luianta Verra [1,2], Bernhard Spitzer [1,3], Nicolas W. Schuck [1,2,4,9] ✉ & Ondrej Zika [1,5,6,7,8,9] ✉

Anxiety has been linked to increased generalisation of threat expectations to perceptually similar stimuli. Such generalisation can arise either from a failure to distinguish threatening from non-threatening stimuli (perceptual mechanism) or from the transfer of learned values between stimuli (value-based mechanism). Yet, how these mechanisms contribute to generalisation remains unclear. Here we assess how participants ($n = 140$) generalise outcome expectancies to perceptually similar stimuli, using personalised stimulus spaces. Computational modelling revealed that individuals differ in the extent to which they generalise value and in the underlying value function. We further found that stronger generalisation in trait anxiety was best explained by greater reliance on value transfer. In this work, we characterise individual differences in the generalisation of aversive stimuli and link stronger generalisation in trait anxiety to preferential reliance on value transfer.

When faced with an unfamiliar situation, we can base our expectations on similar past experiences. If we have been bitten by a dog, for instance, it might seem wise to expect similar-looking dogs to be aggressive as well. But excessive generalisation of aversive experiences can be maladaptive and has been associated with anxiety disorders[1,2] and subclinical trait anxiety[3]. In most lab-based generalisation tasks, a conditioned stimulus (CS+) is paired with an aversive outcome, and during a later generalisation test, participants are presented with perceptually similar generalisation stimuli (GS) that have never been paired with the aversive outcome. These generalisation stimuli typically vary on a perceptual dimension, such as shape or colour. In such stimulus spaces, aversive generalisation can arise from two distinct mechanisms: participants may confuse a generalisation stimulus with the CS+; or they are aware of the perceptual difference but believe that similar stimuli will lead to similar outcomes[4,5]. In this work, we aim to dissociate these mechanisms – perceptual versus value-based generalisation – and ask how trait anxiety affects them.

Previous work has highlighted reduced discriminability between threatening and safe stimuli as one mechanism driving generalisation[4–7]. In scenarios where stimuli vary along a perceptual dimension (e.g., to what degree a new dog resembles the dog that bit us), participants may sometimes perceive a novel stimulus as identical to the previously presented CS+, and treat it as such. Prior work has shown that misidentification increases generalisation responses, depends on the similarity of a stimulus to the CS+, and varies substantially across individuals[5,6,8,9]. These findings highlight the

importance of accounting for perception in generalisation. However, the extent to which perceptual processes contribute to generalisation is a matter of ongoing debate[10,11]. While some studies reported no generalisation when stimuli were well-discriminable, hinting at a major perceptual component in generalisation[12–14], others have found generalisation beyond what can be explained by lack of discriminability alone[5,6,8,15–17]. Perception itself might be impacted by generalisation and learning experiences, highlighting the need to account for perceptual changes over time.

An alternative generalisation mechanism is the active transfer of learned associations based on the similarity of stimuli. If a stimulus is not directly associated with an outcome, we can use the degree of similarity to other stimuli to infer what the outcome probability and intensity (i.e its value) might be. This is an active process where the conditioned value is transferred from one stimulus (CS+) to another (GS) based on their psychological similarity[18,19]. We refer to this process as value-based generalisation, noting that in the current work, value exclusively refers to aversive associations. While psychological similarity can be based on shared perceptual properties (i.e., shape, orientation)[15,20] or on shared conceptual categories or contexts[21], here we specifically focus on perceptual similarities along a single feature dimension. Previous work has manipulated the expected aversive value associated with the CS+ by changing the intensity of the associated aversive stimulus (US) or the probability that it will occur (outcome probability). While generalisation has been shown to increase with threat intensity[22,23], findings on the effect

[1]Max Planck Institute for Human Development, Berlin, Germany. [2]Institute of Psychology, Universität Hamburg, Hamburg, Germany. [3]Faculty of Psychology, Technische Universität Dresden, Dresden, Germany. [4]Max Planck UCL Centre for Computational Psychiatry and Aging Research, Berlin, Germany. [5]Faculty of Psychology and Sports Science, Department of Psychology, Biological and Cognitive Neurosciences, Bielefeld University, Bielefeld, Germany. [6]Centre for Psychiatry Research, Department of Clinical Neuroscience, Karolinska Institutet, & Stockholm Health Care Services, Region Stockholm, Stockholm, Sweden. [7]Department of Clinical Psychology and Psychotherapy, Babeş-Bolyai University, Cluj-Napoca, Romania. [8]School of Psychology, University College Dublin, Dublin, Ireland. [9]These authors contributed equally: Nicolas W. Schuck, Ondrej Zika. ✉e-mail: nicolas.schuck@uni-hamburg.de; ondrej.zika@pm.me

of outcome probability on generalisation have so far been inconclusive[24,25].

A key theoretical and empirical challenge is to dissociate between perceptual and value-based mechanisms of generalisation. One major distinction lies in the fact that only value-based generalisation can account for cases in which generalisation responses deviate from perceptual similarities, for example in cases where participants respond more strongly to generalisation stimuli compared to the CS+ (e.g., Lee et al.[26]). This can for instance reflect participants' subjective impression that the probability of an aversive outcome relates to a particular stimulus dimension and that there could be stimuli that are more aversive than the CS+. If a specific shade of red predicts shock, darker reds may evoke stronger aversive outcome expectations, leading participants to respond more strongly to darker reds than to stimuli most similar to the CS+. This form of generalisation has traditionally been called intensity generalisation[27]. Such monotonically increasing gradients have been reported in a number of studies and typically occur only for some participants[26,28,29]. Monotonic gradients have been mainly reported in human participants[26,27] and are thought to reflect abstract strategies that participants apply to generalise value. Such strategies can be thought of as a form of function learning where stimuli are associated with outcomes based on some inferred function. These generalisation functions can, for instance, be monotonic, reflecting a linear relationship, or Gaussian-shaped, when participants assume that outcomes scale with perceptual similarity. Alternative accounts have associated different generalisation patterns with differences in perceptual discriminability[9,30]. Our paper will investigate which factors influence the occurrence of monotonic or Gaussian-like generalisation gradients and their role in value-based generalisation.

Stronger generalisation of aversive associations is a characteristic marker of anxiety disorders[2,31]. Such proliferation of aversive expectation to perceptually or conceptually similar stimuli leads to an increase in anxiety-evoking cues in the environment[1] and subsequently to diminishing quality of life[32]. Similarly, subclinical trait anxiety has been associated with increased aversive generalisation[3]. Trait anxiety captures individual differences in anxiety symptoms and their severity and is a vulnerability factor for anxiety disorders[33,34]. Whether anxiety impacts perceptual or value-based generalisation processes is unclear. Past work associated trait anxiety with both changes in perceptual discrimination[4,35,36] and with increased value transfer[15]. Our paper will investigate both processes jointly and test how they are impacted by trait anxiety, considering the possibility that trait anxiety may impact different processes in different individuals.

Here we characterise the contribution of perceptual and value-based mechanisms to aversive generalisation using rigorously titrated stimuli and computational modelling. First, we specify two models, conceptualizing perceptual discriminability (perceptual model) and value transfer between similar stimuli (value model). Second, we derive theoretical predictions of behavioural signatures for the two respective mechanisms, and we test these against experimental data in a well-powered online sample ($N = 140$). To be able to dissociate between the two processes, we include personalized stimulus spaces at two levels of discriminability (60% and 80%). To account for sensitivity to outcome uncertainty, we also vary the probability of an aversive outcome (25%, 50%, or 75%), both within-subject. Third, we investigate whether trait anxiety is associated with either perceptual or value-based mechanisms using relative model fit and behavioural metrics.

We hypothesised that individuals will vary in regard to the degree to which their generalisation is driven by value-based processes. Further, we predicted that trait anxiety would be associated with higher rates of generalisation in terms of area under the curve and the estimated model parameter for generalisation strength ($\lambda$). Finally, based on previous reports, we expected that participants would generalise according to different strategies, resulting in different generalisation patterns. Specifically, we expected some individuals to extrapolate the conditioned value in a monotonic way, while others would assume a Gaussian-shaped generalisation function. In an exploratory hypothesis, we predicted that monotonic generalisation will be more common in high trait anxious individuals - since previous work has highlighted the tendency to assume the worst in anxiety.

In this work, we characterise value-based and perceptual contributions to generalisation within a controlled stimulus space and assess their contribution to anxiety-related differences in generalisation. We hypothesize that individuals vary in the extent and the pattern of value generalisation, and that higher trait anxiety is associated with stronger value generalisation.

## Methods

The study complied with and was approved by the internal Ethics committee of the Max Planck Institute for Human Development Berlin (approval number i2023-01). All participants provided informed consent. Data were collected online using Prolific (www.prolific.com), across three experimental sessions. Session 1 included a perceptual titration task and a battery of self-report questionnaires, sessions 2 and 3 contained the main learning and generalisation tasks (see Fig. 1a for overview). The study was not preregistered.

### Participants

One hundred forty participants (93 male, 47 female; mean age: 28, range: 18-40 years) were included in the data set. Participants self-reported their sex. No information about socio-economic status communities of descent or race/ethnicity was collected. Sample size was determined via simulation-based power analysis (simr R package[37,38];), details are provided in the Supplementary Notes 4.

Participants were recruited through Prolific using the following criteria: 95% Prolific approval rate, no current diagnosis of mental illness and no colour blindness. Gender of participants was based on self-report. Out of

**Fig. 1 | Task structure and stimulus space. a** Task structure. Participants first completed a perceptual task where shapes were titrated to determine the step size $\Delta_k$ at 60% or 80% discriminability (low vs high discriminability). $\kappa$ represents the shape's spikiness. During learning, the central shape of the continuum $\kappa = 0.5$ was probabilistically associated with an aversive scream. During generalisation, scream expectancy ratings were collected for generalisation shapes (GS) and for the CS+. **b** Stimulus space for one task condition. The four GSs to either side of the CS+ were chosen based on the titrated $\Delta_k$ for comparable discriminability across participants. Shape spikiness increases from left to right. **c** Task conditions. Reinforcement rates of 25, 50 and 75% were associated with different stimulus colours to create different task conditions (randomised across participants).

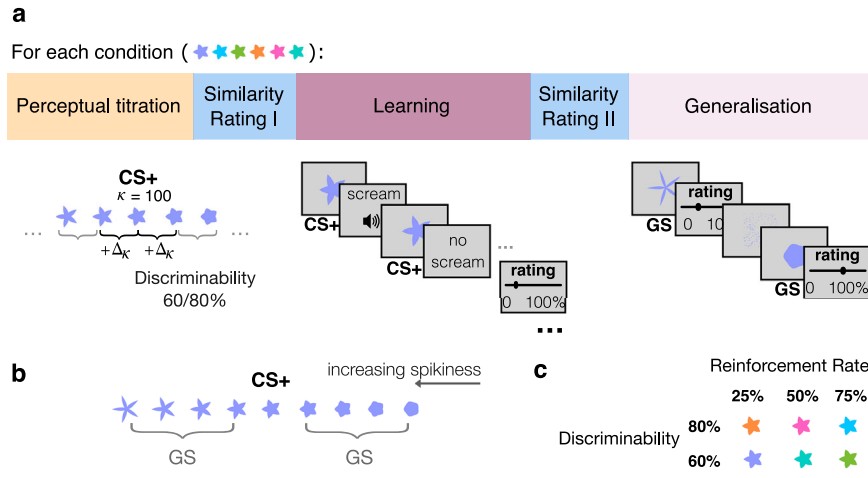

209 participants that completed session one, 33% were not selected to continue with session two (i.e. learning and generalisation tasks), based on predefined accuracy-based performance criteria (see Titration task below). In line with recommendations for exclusion criteria in aversive learning studies, we included data for non-learners in the analysis[39]. No participants were excluded based on performance or attention checks included in the task. Participants were reimbursed 9 GBP for completing session 1 (duration 50 minutes) and 8 GBP for each of session 2 and 3 (40 minutes each).

### Aversive stimuli

Audioclips of female screams were used as unconditioned aversive stimuli for conditioning, as in previous work[40]. These stimuli were sourced from the second version of the International Affective Digitalized Sounds Database (IADS-2, IDE: 277[41]). To standardise sound presentation, participants were required to use headphones during the entire duration of all three sessions. To ensure compliance, participants were required to pass a headphone check task at the start of each session[42]. Preceding the check, participants were presented with a calibration noise sample and asked to adjust the computer volume such that the noise sample was at a comfortable level. Participants were instructed not to adjust the volume during the entire duration of the session. We introduced additional custom audio checks throughout the session to make sure that participants did not turn off the sound. These checks (five per session) consisted of a series of click sounds (1–5) at a low volume that were presented after randomly selected stimulus blocks. Participants were required to indicate the number of clicks they heard. Correct responses were only possible if the audio was on and the volume was not at very low levels. All participants had good performance and passed these auditory sound checks.

### Stimulus space

For each task condition we created a uniquely coloured shape continuum of 9 shapes each (Fig. 1b, c). The shapes were five-fold radially symmetric flower-like shapes adapted from Van Dam et al.[43] generated by connecting two concentric circles of varying relative sizes. This mechanism creates a perceptually continuous space ranging from circular to "spiky". Shape spikiness was defined by the parameter kappa ($\kappa$, range: 0-1), which represents the ratio of the inner and outer radius of the two circles.

For each condition (see below), a stimulus of medium spikiness ($\kappa = 0.5$) was used as the CS+. A 9-item space was then generated by selecting four shapes from each side of this central shape. For each pair of neighbouring shapes, a titration procedure (see below, Fig. 1a) selected a difference in the $\kappa$ parameter to result in a set level of discriminability (80% vs. 60% correct identification of neighbouring shapes as different). To differentiate conditions, the shapes were shown in a distinct colour for each condition (counterbalanced across participants). Shape colours were equidistant, sampled from the CIELAB space with luminance set to 70. Shapes were presented on a grey background (rgb(211, 211, 211)) at a visual angle of 9 deg. The area of all shapes on the continuum was kept constant (i.e., normalised). The shape size for comparable visual angle across participants and sessions was calculated using the virtual-chinrest plugin[44]. This plugin measures participants' viewing distance from the screen by detecting their blind spot location.

### Procedure and design

The experiment consisted of three sessions that had to be completed consecutively or on a maximum of three consecutive days. Participants were told that they would complete a mission in space that involved learning and reporting about a rare species of dangerous screaming space flowers. In each session, participants first gave informed consent and then underwent volume, headphone and screen distance checks (see above), as well as an initial attention check consisting of three questions that could be answered upon careful reading. In session 1 participants completed a titration task, followed by self-report questionnaires. In sessions 2 and 3, participants completed a shorter version of the titration task followed by conditioning, similarity rating and generalisation tasks.

Our main interest was on the aversive conditioning and generalisation tasks. During conditioning, the central stimulus (conditioned stimulus; CS + ) was conditioned using aversive screams (unconditioned stimulus; US). During a subsequent generalisation task, the remaining stimuli (i.e. generalisation stimuli; GSs) were included, and participants' expectancy ratings were collected. Across six conditions, we systematically manipulated the probability that the CS+ would be followed by the scream (outcome probability; 25/50/75% of screams) and the perceptual discriminability of neighbouring stimuli (60% and 80%). Preceding learning, we determined the perceptual discriminability of stimuli to create a personalised stimulus space that was matched in discriminability of stimuli across participants (titration task). To control for possible changes in perception due to learning, we collected similarity ratings for our final stimulus space both before and after learning (similarity rating task). At the end of the experiment, participants were asked about the task, specifically: "Did you worry that potentially there would be an even more dangerous space flower than the super space flower that you have seen screaming?" with the option to indicate "Yes, the very spiky one", "Yes, the very round one", "No" or "other". The experiment was deployed online using the JSPsych JavaScript toolbox[45].

### Titration task

In order to create a personalised stimulus space with comparable discriminability across participants, participants completed a perceptual titration task on session 1. Stimuli were titrated separately for each condition. The goal was to find such perceptual distance $\Delta_\kappa$ which corresponds either to 60% or 80% discriminability. On each trial (n = 40 per condition) participants were presented with two stimuli drawn from the stimulus space continuum (1500 ms). Stimuli from the ends of the continuum ($\kappa < 0.125$ and $\kappa > 0.875$) were not included. Presentation of the first stimulus was followed by a scrambled image (500 ms), a grey screen (varying inter-stimulus interval drawn from truncated exponential distribution [250,1000 ms], M = 500) and a second stimulus (1500 ms). Participants were asked to report if the observed stimuli were identical or different by pressing F or J on the keyboard (max. response time 2000 ms). The response was followed by another scrambled image (500 ms) and a fixation cross (variable ITI; truncated exponential distribution [250,1000 ms], M = 500). Mapping between keys (F/J) and correct response (same/different) was counterbalanced across participants. In half of the trials, the presented shapes differed. On these trials the perceptual distance of the presented stimuli ($\Delta_\kappa$) was adjusted depending on the accuracy of the response. Specifically, $\Delta_\kappa$ was adjusted using a weighted up-down method[46] in steps of 0.005. Depending on the perceptual uncertainty level of a given condition, $\Delta_\kappa$ was titrated to reflect the difference between two stimuli for which participants would correctly identify the shapes as different in 80% (high discriminability and low uncertainty) or 60% (low discriminability and high uncertainty) of times.

As shapes were titrated separately for each condition, $\Delta_{\kappa,low}$ and $\Delta_{\kappa,high}$ were allowed to vary within-participant for conditions with the level of discriminability. This allowed us to account for possible perceptual differences that could result from using different colours. In sessions 2 and 3, shapes were re-titrated in a shorter version of the task (n = 20 trials per condition).

Performance on the titration task in session 1 determined if participants could continue with the main task. Performance criteria were predetermined and required participants to reach an accuracy above chance in all conditions and a difference in final accuracy between high and low perceptual uncertainty conditions greater than 10%. Participants finishing the titration task with $\Delta_\kappa > 0.125$ for one or more conditions were excluded from sessions two and three, as a step size > 0.125 would result in a stimulus space outside the defined range of $\kappa$.

### Learning and generalisation task

Learning and generalisation were blocked per condition. The specific condition order was counterbalanced across participants. During learning, the

central stimulus of each stimulus continuum ($\kappa = 0.5$) was used as the conditioned stimulus (CS + ) and probabilistically associated with a scream (see Fig. 1a: Learning). On each trial, participants were first shown the CS+ (500 ms). The CS+ was followed by the word "feedback" on the screen and a scream on a proportion of trials, depending on the reinforcement rate of the condition (25/50/75%). On unreinforced trials, the "feedback" cue was visible for 1000 ms, matching the duration of the scream. Trials were followed by a variable ITI (drawn from a truncated exponential distribution ([250,1000 ms], M = 500). Participants were asked to observe the trial sequence and to rate how likely it is that the CS+ is followed by a scream halfway and at the end of the trial sequence ($n = 24$). Responses were given using a slider ranging from 0-100%, which was reset to 0 after every trial.

Learning for each condition was followed by the generalisation task (see Fig. 1a: Generalisation). On each trial, one of the 9 stimuli was presented while the participant was asked to indicate how likely it is that this flower will scream (slider rating: 0-100%). A scrambled image (500 ms) followed by a fixation cross (variable ITI drawn from a truncated exponential distribution ([250, 1000 ms], M = 500) was presented in between trials. Each stimulus was presented 4 times in a randomised order. Both the CS+ and the 8 generalisation stimuli were included in the generalisation task. Participants were explicitly informed that no feedback would be given during generalisation. Learning and generalisation for the three conditions were completed in session two, and for the remaining conditions in session three.

### Similarity rating task
To account for possible learning induced changes in perception, participants completed a brief similarity rating task before conditioning (following re-titration) and after conditioning (following generalisation). In this similarity rating task, pairs of neighbouring stimuli from the personalised stimulus space were shown on each trial. Participants were asked to rate how different they perceived the two stimuli from 0 (same) to 100 (completely different) using a slider. A scrambled image (500 ms) followed by a fixation cross (variable ITI drawn from a truncated exponential distribution ([250,1000 ms], M = 500) were presented in between trials. Each stimulus pair was presented twice, resulting in 12 repetitions for each stimulus pair across conditions.

### Questionnaires
Individual differences in trait anxiety were assessed using the State-Trait Inventory for Cognitive and Somatic Anxiety (STICSA[47]). Participants completed questionnaires at the end of session one, using LimeSurvey (Limesurvey GmbH, 2012; Version 3.28.76; www.limesurvey.org).

### Data analyses
Data was analysed using R (Version 4.2.1). The main analyses, including model fitting, are based on aversive probability ratings (0–100%) from learning and generalisation tasks. Analyses for the titration task are based on accuracy and step size, while analyses for the similarity rating task are based on trial-based similarity ratings (0–100%).

### Data pre-processing
To allow for analysis across the two generalisation patterns, a standardisation procedure was performed: First, gradients peaking on the right were flipped horizontally so that all linear gradients would peak on the left. Next, linear ratings were transformed by subtracting stimuli to the left of the CS+ from the mean CS+ of the respective reinforcement rate condition. This transformation preserves the difference of each stimulus from the CS + , a key metric of generalisation. Expectancy ratings were rescaled between 0 and 1 for Beta regression analyses.

### Statistical approach
Statistical analysis of titration responses and parameter estimates was performed using Linear Mixed Models (LMMs, as implemented in the lme4 v1.1-32 R package[48]) and analyses of variance of model results. Analyses of

all ratings were performed using Beta Hierarchical Generalised Linear Mixed Effects Models (GLMMs, as implemented in glmmTMB v1.1.7 R package[49]). Beta regression has been suggested for cases where the variable of interest is continuous and restricted[50] and has been previously used to model ratings[51]. Statistical tests were performed using the Type II Wald $\chi^2$ test as implemented in the car package[52]. Post-hoc tests on regression results are reported using corrected p-values (Holm) and corresponding effect sizes are reported as partial eta squared or Cohen's d with corresponding 95% confidence intervals. Due to difficulties reporting effect sizes for Beta GLMMs, we do not report standardised effect sizes for these models (see glmmTMB package[49]). All tests were two-sided at alpha = 0.05. For all models, we included random slopes for all effects varying within participants[53]. Because LMMs are highly robust to distributional deviations[54], we focused on assessing potential multicollinearity using variance inflation factors, all of which were within acceptable ranges. We also checked for linearity and normality of residuals using visual inspection of QQ and residual plots. For a complete list of packages and versions used in the analysis, see the associated repository.

### Computational modelling
In order to dissociate different mechanisms of generalisation, we employed computational modelling. Models were fitted to trial-by-trial expectancy ratings of each condition of the generalisation task. Ratings of linear gradients peaking on the right were reversed to unify all linear gradients. Final ratings from the learning task were used as starting value estimates for the CS + . Models were fitted using the DIRECT-L global optimisation algorithm[55] as implemented in NLopt in R[56] by minimising the negative log likelihood of the data given a model. We selected this specific optimisation algorithm as it maximised model and parameter recovery results. BIC scores were used to assess model fit as they were shown to recover the models best. Fitting was performed 30 times for each participant and the best-fitting iteration was selected.

Our goal was to model perceptual and value-based generalisation mechanisms across linear and Gaussian-like generalisation patterns. The first model assumes no value generalisation but asks whether generalisation can purely result from perceptual errors, when responses reflect the value learned for the misperceived rather than the shown stimulus. On the other hand, the value model assumes that generalisation arises due to the transfer of value based on the distance of a generalisation stimulus from the CS+, similar to Norbury et al.[15]. We extended existing models by allowing the value to generalise following both a Gaussian-like or a sigmoid generalisation function.

**Perceptual model.** The core assumption of the Perceptual model is that generalisation arises from perceiving the presented stimulus $s$ as another stimulus $q$ (perceived stimulus). The perceptual discriminability of neighbouring stimuli is controlled by a free parameter $\rho \in [0, 1]$ and the probability of misperceiving a stimulus P(q | s) decreases with increasing distance between $q$ and $s$, following a multinomial distribution over stimuli given by

$$P(q|s) = \rho^{d_{|s-q|}} \text{ if } s \neq q$$

$$P(q|s) = 1 - \rho \text{ if } s = q \tag{1}$$

where $d_{|s-q|}$ corresponds to the distance between $q$ and $s$. The probability of correctly perceiving the shown stimulus s is given by $1 - \rho$ Given the assumption of a purely perceptual process, trial-wise predictions $y$ of the model are either 0 when the perceived stimulus q is a generalisation stimulus or the value of the CS + , when the perceived stimulus is the CS + : $y = V_q + \alpha$ (see Fig. 2a). Note that the CS+ value reflects the outcome expectancy each specific participant reported after conditioning, such as to incorporate interindividual differences in learning. The model further

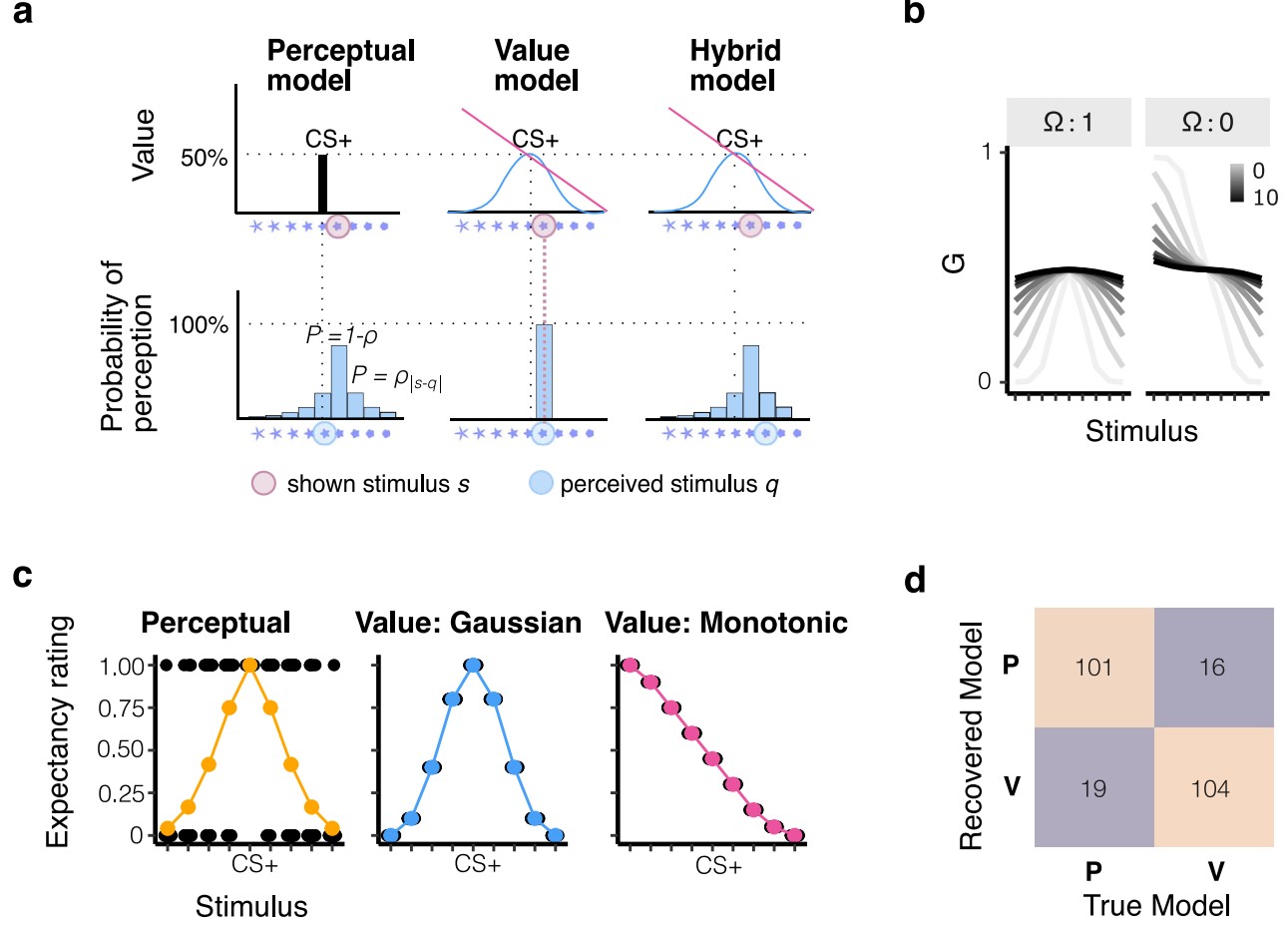

**Fig. 2 | Model predictions. a** Visualization of value-based and perceptual mechanisms across the three models: perceptual, value-based and hybrid models. The top row illustrates the value functions across the three models, which for the perceptual model (top left) is a stick function centred on the CS+ (value directly acquired through learning). The value and hybrid models, in contrast, have Gaussian or linear value functions that assign value to stimuli that were not paired with the aversive outcome during learning (value function parameterized, see **b**). The bottom row shows the probability with which stimuli are misperceived. The perceptual model assumes a probabilistic confusion of each stimulus with neighbouring stimuli, which we empirically derived from the titration task. Shown are the example probabilities of all test stimuli given that GS1 (flower on x-axis) was shown. The value model does not assume any perceptual confusion and, therefore, assigns 100%

probability to the observed stimulus. The hybrid model has the same misperception probability function as the perceptual model. The ratings participants provide are assumed to reflect the value of the perceived stimulus for each model (but see Methods). Predicted response distributions are shown in (**c**). **b** Effects of parameters in the value function G of the value and hybrid models. Each model has a width parameter $\lambda$, which increases generalisation and a shape parameter $\Omega$, which determines whether the function is linear ($\Omega = 0$) or Gaussian-like ($\Omega = 1$). **c** Predicted response distributions of the perceptual and value models (the latter separately for Gaussian-like or monotonic value functions). The stimulus average for both perceptual generalisation and Gaussian-like value generalisation follows a Gaussian-like shape. **d** Model recovery based on BIC (Bayesian Information Criterion) for the perceptual model (P) and value model (V).

accounts for a tendency to over- or under-estimate value across all stimuli by specifying a constant relative offset parameter $\alpha \in [-1, 1]$.

**Value model.** The second model assumes a direct transfer of value based on stimulus similarity (value model). It assumes that the condition-specific value learned for the CS+ ($V_{CS+,c}$) generalises to neighbouring stimuli following a generalisation function G. $V_{CS+}$ represents the individual learned outcome expectancy for the CS+ reported by a given participant after conditioning. The trial-wise predictions $y$ of the model represent the generalised value of the stimulus that was shown, adjusted by an offset $a$.

$$y = a + (V_{CS+} * G_{\Omega}) \tag{2}$$

$G_{\Omega}$ takes a sigmoid-like or gaussian-like shape depending on the strategy parameter $\Omega$ (see Fig. 2b, Eq. 3a/b):

$$\text{if } \Omega = 1 : G_{\text{gaussian}} = \frac{2}{1 + e^{\frac{d_S^2}{\lambda}}} \tag{3a}$$

$$\text{if } \Omega = 0 : \begin{cases} G_{S_{\{1:4\},\text{linear}}} = 1 + \left| 1 - G_{S_{\{1:4\},\text{linear}}} \right| \\ G_{S_{\{1:4\},\text{linear}}} = G_{S_{\{5:9\},\text{gaussian}}} \end{cases} \tag{3b}$$

$d$ indicates the euclidean distance from the CS +, whereas $\lambda \in [0, 10]$ determines the generalisation strength. Higher values of $\lambda$ mean wider generalisation in the Gaussian-like case and less steep slopes in the monotonic case. The upper bound of the generalisation parameter $\lambda$ was selected to capture meaningful differences in data patterns - values above $\lambda=10$ all result in nearly identical flat lines. Lastly, we model a general tendency to over- or under-estimate value by specifying a constant relative offset parameter $\alpha \in [-1, 1]$. We also explored an alternative parametrization, which included trial-by-trial learning. However, this resulted in poorer fits across all conditions, and we therefore excluded it. Predicted response distributions for the perceptual and value model are shown in Fig. 2c.

**Model assessment.** In line with current best practices recommendations[57], we performed model and parameter recovery

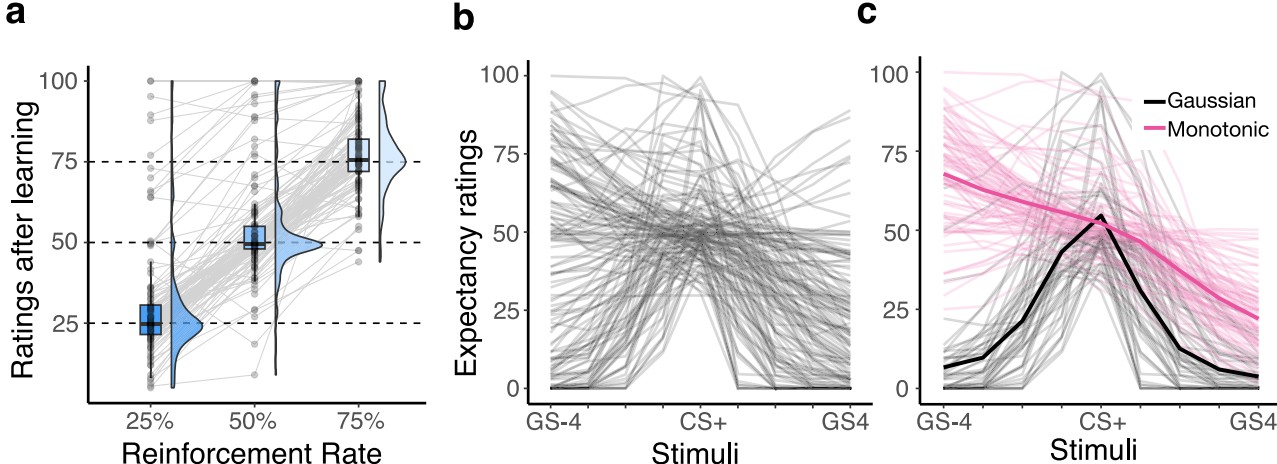

**Fig. 3 | Expectancy ratings for learning and generalisation tasks. a** Expectancy ratings (in %) for the CS+ after learning, split by reinforcement rate conditions. Grey lines represent individual participants (*n* = 140) and horizontal dashed lines represent the target reinforcement rates of 25, 50 and 75%. Each dot represents a participant (*n* = 140 per reinforcement group). The box covers the interquartile range (IQR), mid-line reflects the median, whiskers the +/−1.5 IQR range. **b** Expectancy ratings for all stimuli (GSs and CS+) where individual lines represent the subject average across task conditions (*n* = 140 participants). **c** Transformed expectancy ratings for all stimuli grouped by whether the model assuming Gaussian-like (black) or Monotonic (pink) generalisation fitted best. 75% of participants showed one pattern consistently across conditions; the colour represents the majority pattern within each participant. Bold lines represent the average for each pattern across participants.

procedures in order to establish identifiability across all models and parameters. For both procedures, we generated synthetic data (n = 120) using each model. The parameter range was informed by parameter estimates obtained by fitting the models to participant data; parameters ranging from the 5-95th quantile across participants were used to generate the data (similarly to Zika et al.[58]). To assess parameter recovery, we fitted the model and correlated generative and recovered parameters. All parameters were found to recover well. The correlation between true and recovered parameters was r = 0.75 for the single parameter of the perceptual model and r > 0.9 for all parameters of the value model. Importantly, the generalisation parameter $\lambda$ was found to recover well across the entire range of values ($r_s$= 0.9, p < 0.001 (see Supplementary Fig. 4 and 5 for an overview of parameter recovery and parameter correlations). Subsequently, model recovery analysis was performed by fitting all models to the generated data and assessing how reliably the best-fitting model (in terms of BIC) corresponded to the model used to generate the data. Models were found to recover well; the value model recovered correctly in 104/120 instances (86.67%) while the perceptual model recovered correctly in 101/120 instances (84.17%, Fig. 2d).

### Reporting summary

Further information on research design is available in the Nature Portfolio Reporting Summary linked to this article.

## Results

Our main goal was to study how generalisation of expectations to perceptually similar stimuli can arise from perceptual and value-based mechanisms. 140 participants (47 female, mean age: 28 years) first learned to probabilistically associate a flower-like shape (conditioned stimulus, CS + ) with an aversive scream (US, learning phase). In a second step, ratings for eight stimuli varying in similarity to the conditioned stimulus were collected (generalisation phase). Prior to the main task, the discriminability of these stimuli was titrated in order to create personalised stimulus spaces matched in perceived similarity across participants. Stimulus similarity ratings were additionally collected before and after conditioning to track possible learning-induced perceptual changes (see Fig. 1a for task overview). Across a set of 6 blocks, we varied the reinforcement rate of the CS+ (3 levels; 25%, 50% and 75% scream probability) and the discriminability level (2 levels; 60% and 80%). Each block contained both a learning and generalisation phase. The colour of the stimulus set varied between the 6 blocks (Fig. 1c).

Results from the perceptual titration show that participants' discrimination performance was titrated as intended, i.e. there was no evidence for a difference between participant performance and the target accuracies in the 60% (mean: 60.12%, sd: 7.53%; t(139) = 0.19, p = 0.852, d = 0.02, CI = [−0.15,0.18]) and 80% conditions (mean: 79.88, sd = 6.92%, t(139) = −0.2, p = 0.839, d = −0.02, CI = [−0.18,0.15]). Participants also approximated the true reinforcement levels well during learning (mean$_{25\%}$ = 0.31, sd$_{25\%}$ = 0.18; mean$_{50\%}$=0.54, sd$_{50\%}$ = 0.14; mean$_{75\%}$ = 0.77, sd$_{75\%}$ = 0.11), although they had a tendency to overestimate the true level in all three conditions (condition$_{25\%}$: t(138) = 4.10, p < 0.001, d = 0.34, CI = [0.16,0.51]; condition$_{50\%}$: t(139) = 3.51, p < 0.001, d = 0.3, CI = [0.13,0.47]; condition$_{75\%}$: t(137) = 2.03, p = 0.04, d = 0.15, CI = [0.02, 0.32]; Fig. 3a; see Supplementary Notes 1. Next, we analysed participants' ratings during the generalisation phase for stimuli not seen during learning and which differed from the original stimulus in shape spikiness. A large proportion of ratings (82%) for generalisation stimuli were greater than 0. This could suggest either orienting responses or other sources of response noise, or transfer of aversive outcome expectations to generalisation stimuli. The modelling reported below will test the latter possibility.

Visual inspection of gradients revealed that participants generalised aversive expectations to previously unseen stimuli using two distinct generalisation patterns (Fig. 3b), in line with previous reports[26,28]. A first subgroup rated stimuli in a monotonically increasing or decreasing manner along the roundness-spikiness continuum. The most extreme stimuli (spiky/round) were rated as most likely to result in the aversive outcome. The second subgroup exhibited a Gaussian-like generalisation pattern, i.e. expectancy ratings decreased in both directions as a function of perceived similarity to CS+ (Fig. 3c). A model-based analysis (see below) estimated the proportion of the monotonic and Gaussian-like gradients to be 55.12% and 44.88% respectively (see Supplementary Fig. 3 for classification of individual gradients per condition).

Overall, outcome expectancy ratings were higher for Monotonic compared to Gaussian-like gradients, mean$_{Monotonic}$: 47.3, mean$_{Gaussian}$: 26.3, t(787) = -17.23, p < 0.001, d = 1.2, CI = [1.04,1.34,].

### Dissociating mechanisms based on trial-level responses

To capture differences in the underlying generalisation mechanisms (perceptual vs value-based), and in the generalisation pattern (monotonic vs Gaussian-like), we developed two computational models. Both models assume that participants' rating responses reflect their underlying outcome

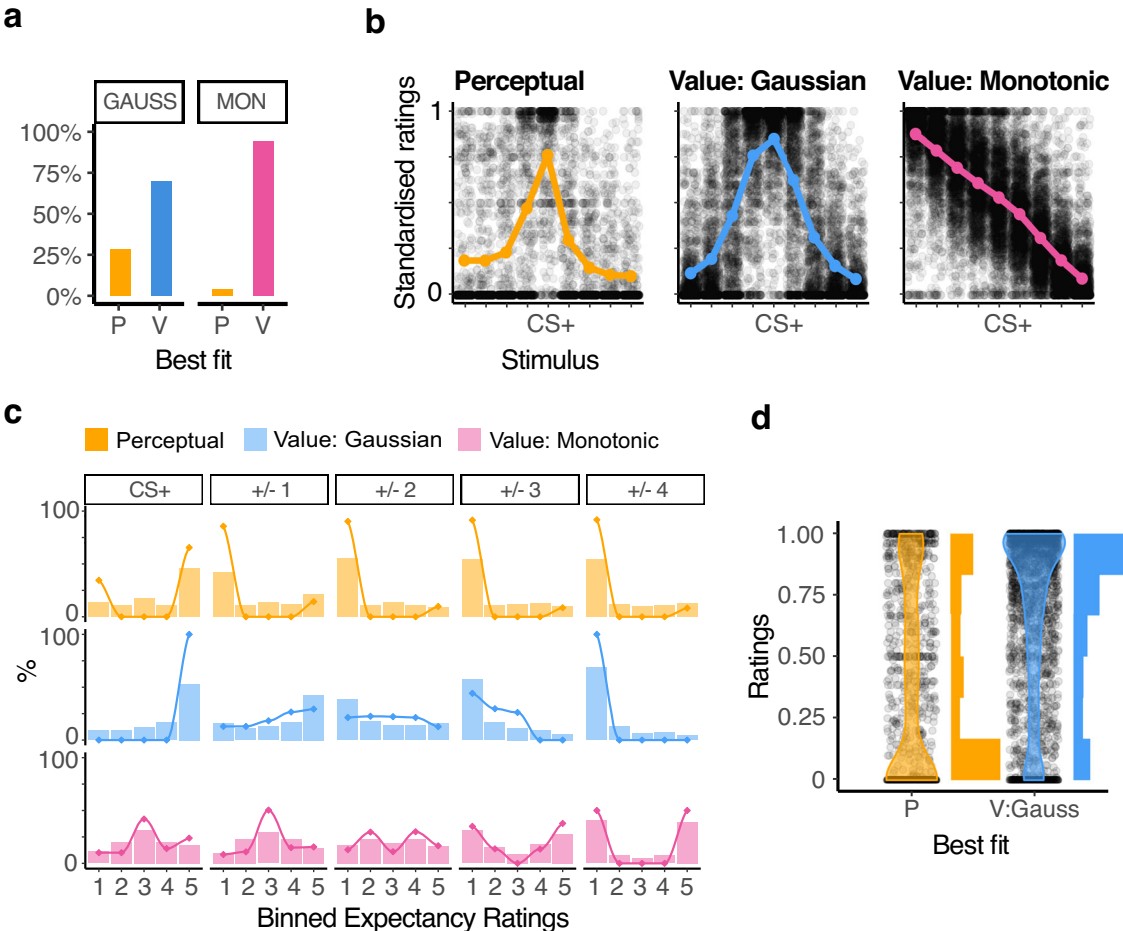

**Fig. 4 | Modelling results. a** Best fitting model (perceptual model- P, value model - V) in % split by generalisation pattern (Gaussian-like vs. Monotonic) as classified by $\Omega$. **b** Standardised responses for each stimulus split by model fit and pattern (gradients classified as perceptual $n = 128$, value gaussian: 265, value monotonic: 447). Lines represent the stimulus average. **c** Histograms of response for each stimulus, again split by best model fit and pattern. Responses are standardised to allow plotting across reinforcement rate conditions. Value-based monotonic and Gaussian-like gradients are plotted on the same scale; however, the range of the responses differs for the groups, given that ratings for monotonic gradients can be larger than the CS+ in about half of the stimuli. Lines show model predictions for the respective models. Simulated data was analysed and binned the same way as responses, lines were smoothed for plotting **d** Response distribution for the GSs adjacent to the CS+ (distance +/-1) for gradients classified perceptual ($n = 128$) or value gaussian ($n = 265$). Responses for gradients best fit by the perceptual model approximate a binary distribution of 0 and 1. Responses for gradients best fit by the value model show a gradual distribution.

expectancies. The perceptual model assumes that generalisation occurs as a consequence of impoverished discriminability. In this model, the perceived stimulus on a given trial is assumed to be a single sample from a probability distribution centred on the presented stimulus. The probability of perceiving a particular stimulus decreases with its distance from the presented one, scaled by an uncertainty parameter ($\rho$). Two predictions arise from this account: First, rating response distributions to generalisation stimuli should be bi-modal, reflecting that the same stimulus was sometimes perceived as different from the CS+, resulting in a rating of 0%, and on other occasions confused with the CS+, resulting in a rating that matches CS+ expectations (Fig. 2a). Under this hypothesis, stimuli close to the CS+ are expected to be misperceived as the CS+ more frequently which, on average, may give rise to a Gaussian-like pattern (Fig. 2c, *left*). We note that other factors can, in principle, result in a binary response pattern, such as participants employing a binary response strategy, an alternative explanation we tested in the Supplementary Notes 6 (see below). Second, perceptual mechanisms cannot account for monotonic generalisation patterns, since the confusability with the CS+ decreases with perceptual distance. Our model does not assume systematic shifts in perception or memory, such as misremembering and misidentifying the most extreme stimulus as the CS+. Hence, perceptual generalisers are expected to show Gaussian-like response patterns when averaged, but bi-modal distributions on a trial wise level.

The second model reflects a value account that assumes non-zero outcome expectations for unseen stimuli arise due to transfer of value from CS+ to GSs. The key distinction from the perceptual model is that the ratings themselves, not just their average, will gradually change with the perceived distance from the CS+. Furthermore, in line with previous work, we assume that participants draw from a set of prior functions that determine how they extrapolate the observed CS+[59], and specifically consider the Gaussian-like or monotonic shape as possible kernels in this set of priors. The shape of the generalisation function G, is governed by a free parameter $\Omega$ (1=Gaussian-like; 0=monotonic). This allows us to capture differences between monotonic and Gaussian-like generalisation patterns that may occur during value-based transfer (Fig. 2b, c *middle* and *right*). Independent of its shape, G also parametrizes the strength of generalisation (parameter $\lambda$) and a general offset (parameter $\alpha$). Model recovery showed that models recovered well; the perceptual model recovered correctly in 84.17% of cases, the value model was correctly identified in 86.67% of cases (see Fig. 2d).

Applying these models to participants' data across conditions revealed a better fit of the value model compared to the perceptual account, as shown by the Bayesian Information Criterion (BIC: 215739 vs 235109). Assessing the best model for each participant and condition revealed that some gradients were better fitted by the perceptual model (15.24% of overall gradients, Fig. 4a). Among those gradients best fit by the value model (84.76%),

37.22% were classified as Gaussian-like and 62.78% as monotonic using the model parameter $\Omega$. Generalisation patterns were relatively consistent within participants: 75% of participants showed the same type of generalisation pattern consistently across all six task conditions. In post-task feedback, 76.47% of participants with the majority of gradients classified as monotonic affirmed that they worried that there would potentially be an even more dangerous stimulus than the CS+ compared to 20.63% in the gaussian case (feedback was not recorded for 9 participants). We additionally tested a hybrid model combining value generalisation (value model) with misperception of stimuli (perceptual model) as illustrated in Fig. 2a. Model comparison favoured the value-based (76.90%) over the hybrid (8.1%) and perceptual (15.0%) models, suggesting that the value component explains most variability in behaviour. We further tested a binary choice model to test if the binary response pattern observed in our data could reflect a binary decision rule. 11.67% of gradients were best fit by this model, out of which only 3.06% were previously best fit by the perceptual model (see Supplementary Notes 6).

We next confirmed that model predictions about trial-wise rating distributions were evident in the data. Splitting participants according to the model and shape that best fit their behaviour, we asked whether the non-averaged ratings of perceptual generalisers had a bimodal distribution, while the ratings of value-based generalisers were distributed uni-modally. In order to assess all outcome probability conditions jointly, we rescaled the responses within each outcome probability condition to range between 0 and 1. This analysis revealed that perceptual generalisers indeed had a tendency to rate generalisation stimuli as either 0 or the CS+ level, exhibiting a marked bimodality. Conversely, response patterns for the value-based participants changed gradually based on the distance from CS+: In Gaussian-shaped gradients, this was expressed as a gradual distance-based decrease for stimuli on both sides of the CS+, while in monotonic gradients, ratings gradually increased or decreased depending on whether stimuli were on the left or right side of the continuum relative to the CS+ (Fig. 4b). These distinct distributions were also reflected in response histograms of the Gaussian-like shapes (both perceptual and value-based, Fig. 4c).

Focusing on Gaussian-shaped gradients across both models, the difference between perceptual versus value-based model fits was most evident in responses to GSs neighbouring CS+. The perceptual group showed a bimodal response distribution while Gaussian-like value generalisers showed a distribution with peak at the CS+ (Fig. 4d). With increasing distance from the CS+, responses in the perceptual group were mostly 0. In the Gaussian-like value group, most responses were non-zero and gradually decreased with distance from the CS+. Similarly, ratings in the monotonic value group gradually increased and decreased with distance.

### Measures of Generalisation
Next, we analysed parameter estimates of the value model (see Supplementary Table 1 for overview). We first focussed on the width parameter $\lambda$ of the generalisation function G, and confirmed that it relates to individual generalisation gradients. To obtain a measure of generalisation strength, we estimated the area under the curve (AUC) for each individual and condition using the trapezoidal rule for numerical integration. We then fit a linear mixed effects model (LMM) to test for a relationship between the AUC and the model derived generalisation parameter $\lambda$. This analysis revealed a significant positive effect of AUC on $\lambda$, $\beta=12.78$, $\chi^2(1) = 95.78$, $p < 0.001$, $\eta_p^2 = 0.11$, CI = [0.07,0.16], indicating that higher values of $\lambda$ were associated with higher AUC, as expected.

We next assessed how the generalisation strength parameter $\lambda$ was impacted by our experimental manipulations, and the generalisation pattern as classified by $\Omega$. Modelling $\lambda$ as a function of the generalisation pattern, reinforcement rate condition, perceptual discriminability condition, and the interaction of the experimental manipulations with the generalisation pattern revealed a main effect of perceptual discriminability, reflecting that higher discriminability (the 80% condition) was associated with lower $\lambda$ values, compared to the low discriminability condition (60%), $\chi^2(1) = 32.32$, $p < 0.001$, $\eta_p^2 = 0.18$, CI = [0.08,0.3], lambda$_{high-low}$:

$t(139) = -5.52$, $p < 0.001$, $\eta_p^2 = 0.18$, CI = [0.08,0.29]. The model further showed an interaction of reinforcement rate and generalisation pattern, $\chi^2(2) = 38.72$, $p < 0.001$, $\eta_p^2 = 0.11$, CI = [0.05,0.17]: higher reinforcement rate was associated with higher values of $\lambda$ in monotonic but not Gaussian-like gradients, monotonic$_{low-mid}$: $t(166) = -4.701$, $p < 0.001$, $\eta_p^2 = 0.24$, CI = [0.15,0.34], monotonic$_{high-mid}$: $t(167) = 4.67$, $p < 0.001$, $\eta_p^2 = 0.12$, CI = [0.04,0.21]. These results show that the value associated with the CS+ impacts generalisation differently depending on the generalisation pattern.

We next assessed if the experimental manipulations impacted the offset parameter $\alpha$, that allows values across the stimulus space to overall increase or decrease. A model with $\alpha$ as function of the fixed effects and interaction of the reinforcement rate and discriminability conditions revealed a main effect of reinforcement rate, $\chi^2(2) = 38.74$, $p < 0.001$, $\eta_p^2 = 0.20$, CI = [0.09,0.30], $\alpha$ was more negative with increasing reinforcement rate, RR75%-RR50%: $t(139) = -3.57$, $p < 0.001$, $\eta_p^2 = 0.21$, CI = [0.1,0.33]; RR25%-RR50%: $t(139) = 2.8$, $p < 0.001$, $\eta_p^2 = 0.08$, CI = [0.02,0.18].

In generalisation functions, peak and width are co-dependent; peak differences arising from reinforcement conditions or $\alpha$ can therefore impact the interpretation of $\lambda$. While higher reinforcement rates were associated with increased $\lambda$, we next tested how $\alpha$ relates to $\lambda$. Parameter recovery had revealed that the parameters were uniquely identifiable and thus not correlated by default. A LMM model with $\lambda$ as the dependent variable and $\alpha$, the reinforcement condition and their interaction as fixed effects showed a negative association of $\alpha$ and $\lambda$, $\beta=-3.55$, $\chi^2(1) = 99.65$, $p < 0.001$,$\eta_p^2 = 0.13$, CI = [0.09,0.18]. Together, this suggests that while reinforcement rate is associated with increased $\lambda$, larger $\alpha$ is associated with decreased $\lambda$.

### Trait anxiety effects on generalisation
Finally, we turned to our main question, namely whether differences in trait anxiety were associated with differences in generalisation. Trait anxiety was assessed using the State-Trait Inventory for Cognitive and Somatic Anxiety (STICSA mean: 29.34, range: 21-84; see Fig. 5a).

We first asked how trait anxiety affected participants' rating behaviour. A first analysis showed that there was a global association between trait anxiety and the average magnitude of shock expectations across all generalisation stimuli, pearson's r = 0.11, CI = [0.05, 0.17], $p < 0.001$ and further, that trait anxiety was correlated positively with the estimated area under the curve (AUC), Pearson's r = 0.12, CI = [0.06, 0.19], $p < 0.001$. Using a Beta GLMM to jointly test relations between expectancy ratings and trait anxiety, distance of a stimulus from the CS+, reinforcement rate and discriminability, revealed a positive interaction of anxiety and distance, $\beta= 0.03$, $\chi^2(1) = 25.25$, $p < 0.001$, indicating that the effect of anxiety on expectancy ratings increased with increasing distance from the CS+, Fig. 5b. While this model included all main effects and interactions (as well as nuisance variables for monotonic/gaussian-like response patterns, see below), we did not find statistically significant evidence that trait anxiety interacted with the experimental manipulation of the reinforcement rate, $\chi^2(2) = 2.45$,p = 0.293, or discriminability $\chi^2(1) = 0.33$, p = 0.565. We next assessed the association between the model derived generalisation parameter $\lambda$ and trait anxiety and found no statistically significant main effect of trait anxiety on generalisation width, $\beta = 0.004$, $\chi^2(1) = 0.00$, p = 0.98, $\eta_p^2 = 0.00$, CI = [0.00,0.00]. As reinforcement rate, discriminability and offset were found to impact $\lambda$, we specified a model of $\lambda$ including these variables and their interaction with trait anxiety. This model revealed a significant 3-way interaction of trait anxiety, $\alpha$ and reinforcement rate, $\chi^2(2) = 14.12$, $p < 0.001$, $\eta_p^2 = 0.02$, CI = [0.00,0.05], suggesting that trait anxiety impacts $\lambda$ differently depending on $\alpha$ and the reinforcement condition. Figure 5c illustrates this relationship, showing the effect of trait anxiety on generalisation strength after accounting for other variables that impact $\lambda$.

While the effects of anxiety on generalisation width shown above are known, whether this reflects value-based or perceptual mechanisms is less understood. A tendency towards value or perceptual generalisation should be reflected in improved model fits of one model over the other. To answer

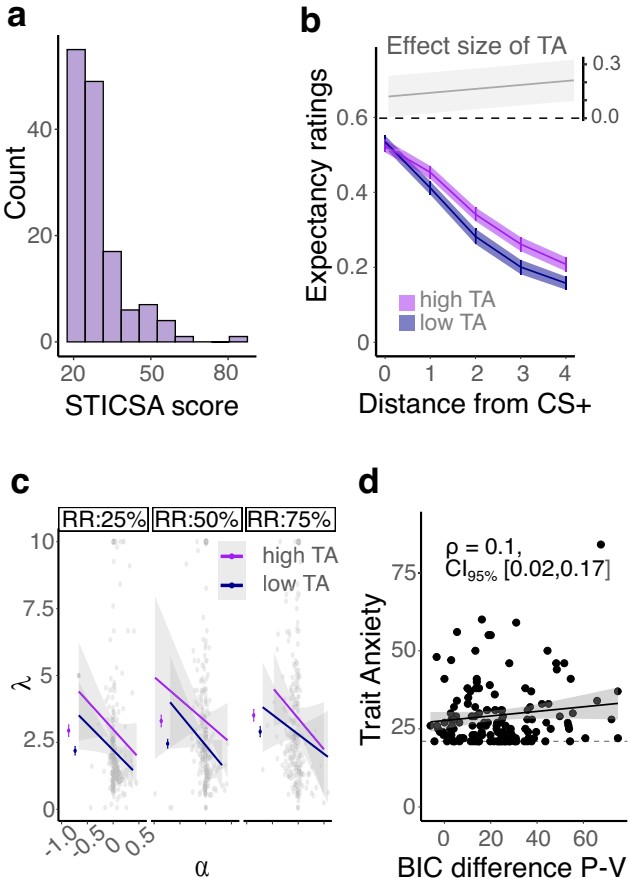

**Fig. 5 | Trait anxiety effects on generalisation. a** Histogram of STICSA scores ($n = 140$ participants). **b** Transformed expectancy ratings by trait anxiety (TA) split by median into high vs. low trait anxious participants. Lines show mean ratings per anxiety group ($n = 70$ per group), shaded areas show standard errors of the mean. The figure insert shows the effect of trait anxiety on generalisation ratings, which increases with increasing distance of a stimulus from the CS + , $\beta= 0.03$, $\chi^2(1) = 25.25, p < 0.001$. **c** Association of model parameters λ, α and the experimental reinforcement rate manipulation and their relation to anxiety. Each dot represents one participant ($n = 140$), the lines represent the association of α with λ by the anxiety group (median split into high and low). The figure inserted to the left of the scatterplots shows the mean lambda by anxiety group. Bars represent the standard error of the mean. **d** Relationship of trait anxiety and relative model fit, where trait anxiety was associated with better relative fit of the value model across generalisation patterns, $\beta= 3.37$, $\chi^2(1) = 5.24, p = 0.022, \eta_p^2 = 0.006[0.0,0.05]$. Each dot represents one participant ($n = 140$), the relative model fit was, however, based on fits per task condition ($n = 840$). We show the correlation of trait anxiety and relative model fit with the 95% confidence intervals obtained through bootstrapping. The dashed line represents the lowest possible STICSA value.

this question, we related the difference in model fit to fixed effects and interactions of trait anxiety, reinforcement rate and discriminability. The model identified a positive relationship between trait anxiety and relative model fit $\beta= 3.37$, $\chi^2(1) = 5.24$, p = 0.022, $\eta_p^2 = 0.006$, CI = [0.00,0.05], indicating that trait anxiety was positively associated with a tendency towards better fit of the value model (Fig. 5d). This relationship was also evident in a simple correlation analysis (Pearson's r = 0.1; bootstrapped 95% CI = [0.02,0.18]). We found no statistically significant moderation of this effect by reinforcement rate, $\chi^2(1) = 1.2$, p = 0.27, $\eta_p^2 = 0.001$, CI = [0.00,0.02], or discriminability condition, $\chi^2(1) = 0.24$, p = 0.62, $\eta_p^2 = 0.0006$, CI = [0.00,0.01]. High trait anxiety levels were also associated with model fit consistency across task conditions, as quantified by comparing trait anxiety of participants consistently best fit by the same model versus those fitted best by different models across conditions, t(137) = -2.09,

p = 0.039, d = 0.34, CI = [0,0.68]. This indicates a tendency for mechanism consistency in high trait anxiety.

Finally, we assessed if trait anxiety was associated with the use of monotonic versus Gaussian-like generalisation in participants best fitted by the value model. A logistic model with the fitted pattern as the dependent variable showed no statistically significant effect of trait anxiety $\chi^2(1) = 1.31$, $p = 0.25$. Together, these findings suggest that stronger generalisation in trait anxiety is associated with a tendency to generalise value. This effect seems to be independent of the specific value of the conditioned stimulus or the generalisation pattern used.

### Perceived similarity, learning and generalisation
Finally, we explored whether the generalisation mechanism (perceptual/value-based) or the generalisation pattern (monotonic/Gaussian-like) was associated with how stimuli were learned or perceived. We first tested for associations of stimulus discriminability (titration task) and stimulus perception (similarity rating task) with generalisation mechanisms or patterns and found no statistically significant association with accuracy in the titration task or rated perceived similarity of stimuli (see Supplementary Notes 1). We found an association between differences in CS+ expectancy after learning and generalisation mechanism, $\chi^2(1) = 8.06$, $p < 0.01$, $\eta_p^2 = 0.09$, CI = [0.0,0.25], with post-hoc tests showing higher CS+ ratings after learning in participants better fit by the value model $mean_{value}$: 0.54, $mean_{perc}$: 0.50, value-perc: z = 2.84, $p = 0.004$, d = 0.08, CI = [0.01,0.15]. When comparing CS+ values after learning for participants best fit by the value model, we further found and effect of pattern, $\chi^2(1) = 5.47$, p < 0.019, $\eta_p^2 = 0.09$, CI = [0.0,0.24], with post-hoc tests indicating that a Gaussian pattern was associated with higher CS+ values after learning, $mean_{gauss}$: 0.61, $mean_{monotonic}$: 0.5, gaussian-monotonic: z = 2.34, p = 0.019, d = 0.09, CI = [0.01,0.17]. Exploratory analysis further revealed a negative association of learning speed and generalisation strength (see Supplementary Notes 1 and 6). Finally, we asked whether learning impacted perceived similarity of stimuli. Analysis of similarity ratings collected before and after learning revealed that participants perceived stimuli as more similar after conditioning, $mean_{pre}$: 0.41, $mean_{post}$: 0.37, pre-post: z = 6.09, p < 0.001, d = 0.08, CI = [0.05,0.10]. Interestingly, we found no statistically significant effect of distance of a stimulus from the CS+ and this learning-related effect, $\chi^2(1) = 0.86$, p = 0.354.

To assess if differences in stimulus perception or learning were associated with trait anxiety, we modelled final accuracy in the titration task, rated similarity and outcome expectancy for the CS+ as a function of trait anxiety. These models found no statistically significant association of any of the measures with trait anxiety (see Supplementary Notes 2).

### Discussion
We investigated how perceptual and value-based mechanisms contribute to generalisation of threat expectations and whether trait anxiety is preferentially associated with either mechanism. Based on outcome expectancy ratings for stimuli varying in similarity to a conditioned stimulus, we show that perceptual and value-based generalisation can be dissociated through trial-level response distributions. Computational modelling revealed variability in participants' reliance on different mechanisms, with some gradients being best explained by lack of discriminability alone. We show that trait anxiety is linked to stronger generalisation to stimuli farthest from the CS + , along with a tendency to generalise value.

Our results show that in perceptually continuous spaces participants generalise either in a monotonic or Gaussian-like manner, in line with previous studies reporting different generalisation patterns[26,28]. We show that Gaussian-like gradients can arise from two distinct underlying response distributions characteristic of perceptual and value-based generalisation. Specifically, participants best fit by the perceptual model show a bimodal response distribution while participants best fit by the value model show a more uniform distribution of responses. This finding provides behavioural and computational evidence for two distinct mechanisms of generalisation and contributes to our understanding of generalisation in at least two ways.

First, participants differed in how likely they were to rely on value generalisation, as was reflected by the relative model fits of the perceptual and value-based models. Dissociating value-based from perceptual mechanisms is particularly challenging in Gaussian-shaped generalisation patterns. Previous literature disagrees on the contribution of perceptual and value-based mechanisms to generalisation. Here we show that poor discriminability alone could explain generalisation in about 15.24% of gradients, suggesting that it is likely not the sole explanation for generalisation as has been argued in previous work[13,14]. Our data suggest that generalisation to perceptually similar stimuli further relies on value-generalisation, consistent with previous work modelling perceptual differences in generalisation[15,17]. While our modelling approach focuses on dissociating behavioural signatures of perceptual and value-based generalisation, our models can be combined to account for joint value generalisation and misperception; an example is provided in the Supplementary Notes 6. Note that our perceptual and value models both assume that participants' value beliefs are directly expressed in their ratings. To test this assumption, we fitted an alternative model that includes a transformation of value estimates in a manner that can result in binary ratings. This model, however, fitted the data less well.

Second, by titrating generalisation stimuli to two discriminability levels, we show that the level of discriminability relates to the amount of apparent generalisation. This was evident from higher values of $\lambda$ in the low discriminability compared to the high discriminability conditions, indicating that within-person manipulation of discriminability resulted in differences in generalisation. Perceived similarity ratings confirmed our perceptual manipulation - stimuli titrated to 60% discriminability were perceived as more similar than those titrated to 80% discriminability. Stimuli in many generalisation experiments are based on physical stimulus features rather than on subjects' perception[4,7] which often makes it difficult to control for discriminability. Recent modelling work accounting for stimulus perception has shown how similar generalisation tendencies can produce different responses depending on how stimuli were perceived[17]. Here we manipulate stimulus discriminability and show how discriminability influences generalisation gradients, adding to findings associating an individuals' perceptual acuity to the degree of generalisation. Because prior studies have used stimulus spaces with differing overall stimulus similarity, comparisons across studies are difficult; we would however expect that spaces titrated to lower discriminability levels would result in overall higher misperception.

We show that discriminability relates to the amount of generalisation, and that despite holding discriminability constant, one quarter of participants showed different generalisation patterns across task conditions. This suggests that individual differences in generalisation vary beyond what can be explained by perceptual differences alone. It further contrasts recent work suggesting perceptual differences as a pathway to understanding individual differences in generalisation patterns[9] such as work relating more accurate CS+ identification with a tendency for similarity based generalisation[30].

While previous work has linked trait anxiety to increased aversive generalisation[3], a number of critical insights are lacking. First, it is unclear whether trait anxiety impacts perceptual or value-based processes. This distinction is essential for clarifying mechanisms that drive increased generalisation in anxiety and consequently for developing targeted interventions. Here, we show that trait anxiety was associated with a better relative fit of the value-based model compared to the perceptual model, indicating that anxiety is associated with a tendency to generalise value. This aligns with findings associating trait anxiety with value generalisation[15] and provides a pathway into understanding what drives anxiety-related differences in generalisation. It challenges some previous findings arguing that anxiety-related differences in generalisation stem from how stimuli are perceived[35]. Taken together, this evidence suggests that trait anxiety is associated with a tendency to generalise value to stimuli that are perceptually similar. While further research is needed, this finding points to the potential relevance of clinical interventions based on expectation management. While our findings apply to perceptually similar stimuli, further research is needed to understand how anxiety affects generalisation to conceptually or semantically related stimuli.

The current work focuses on mechanisms that explain differences in reaction to threat and relates individual differences to trait anxiety. High self-reported trait anxiety is a risk factor for the development of clinical anxiety disorders[33,34]. Computational models targeting trait-anxiety related differences in threat processing are a valuable resource for understanding factors that contribute to the aetiology and maintenance of anxiety disorders[60]. Our approach sets the basis for the measurement of generalisation and offers a starting point for identifying targets for clinical interventions (e.g., in precision psychiatry). Further work is warranted to ensure that findings based on trait anxiety apply to clinical populations, keeping in mind that trait anxiety is not a sharply defined construct and overlaps with related traits such as neuroticism, chronic worry and negative affect[61].

Second, from previous work, it is unclear whether anxiety impacts stimuli closest or furthest to the CS+. Although perceptual similarity between the CS+ and generalisation stimuli decreases along the generalisation gradient, the ambiguity related to the outcome increases. We show that trait anxiety was associated with broader and less discriminative gradients, which translated to higher outcome expectancy ratings in stimuli most dissimilar to the CS+. Similarly, previous work using different stimulus shapes has found broader generalisation gradients in high trait anxiety[62]. Our findings partly contradict work relating trait anxiety to overprediction of outcome expectancies at the CS+[63] and highlights sensitivity of anxious individuals to ambiguous and uncertainty stimuli. This observation aligns with previous work highlighting the role of unpredictability and uncertainty (as well as intolerance of) as a key characteristic of anxiety[64,65]. Together, these findings suggest that trait anxiety is linked to stronger generalisation toward stimuli that are increasingly distinct from the CS+, corresponding to stimuli with the highest ambiguity surrounding them.

Third, it is unclear whether anxiety relates to monotonic or Gaussian-like generalisation patterns. While Gaussian-like generalisation is rooted in perceptual similarities, extrapolating in a monotonic way requires the assumption of an implicit generalisation rule. One exploratory analysis was whether anxious individuals will have a tendency to apply such rule-based patterns in order to reduce their subjective uncertainty[58]. We find no statistically significant association between trait anxiety and a specific pattern, but rather that trait anxiety impacted generalisation across patterns - indicating a general tendency to generalise value. This aligns with work relating trait anxiety to ambiguity about the specific underlying rule rather than to a specific pattern[62]. It partly contrasts previous evidence relating specific pattern characteristics to anxiety[66], though anxiety-related differences in previous work emerged already at acquisition.

Finally, we show that outcome probability positively influenced generalisation strength as shown by higher values of $\lambda$ with increasing outcome probability. These findings suggest that the optimal strength of generalisation depends on the current threat level, making it hard to define where optimal generalisation becomes maladaptive. This effect was however, inconsistent across generalisation patterns, calling for a more detailed investigation into the effect of outcome probability and value magnitude on generalisation for specific patterns.

## Limitations

While this study provides valuable insights into perceptual and value-based contributions to generalisation, certain methodological constraints must be acknowledged. Perceived similarity ratings of neighbouring stimulus pairs suggested the stimulus space is not fully perceptually linear, specifical,y stimuli at the edges of the space seemed to be perceived as more distinct. This contrasts findings of Van Dam et al.[43] who found the stimulus space to be perceptually linear. It is unclear if perceived similarity directly translates to discriminability of stimuli, providing a possible explanation for the diverging results. Further, emotional events like aversive conditioning are thought to affect stimulus perception, although evidence is mixed. Comparing similarity ratings before and after conditioning, we found that

participants rated stimuli as overall more similar after learning. This aligns with a recent publication reporting no specific learning-induced changes in perceptual acuity[67] and likely reflects a fatigue effect rather than an effect of learning. In addition, we acknowledge that other variables might have introduced noise to participants' responses, such as poor memory or attention. Recent work has shown that participants can systematically misremember which stimulus was associated with a scream[9,30] and that such memory bias can account for differences in generalisation pattern. While we did not observe systematic shifts in CS+ memory, we cannot dissociate noise due to poor memory or attention effects from perception in the current task design.

Finally, we acknowledge that our experiment did not systematically manipulate factors that could impact the balance of perceptual and value based generalisation, and hence relies on describing pre-existing differences between participants of unknown origin. To strengthen mechanistic insights, future work should therefore seek to experimentally manipulate process-relevant variables. In this context, we note that our participants were instructed to rate how likely a scream is, which could have led to a higher prevalence of value-based responding. Such a question, however, can best be solved by experimental manipulation.

Although we controlled for many possible confounding variables, we did not find an explanation for the use of a specific generalisation pattern. Previous reports have found a similar proportion of monotonic and Gaussian-like patterns in their sample and have further shown that instructional manipulations can influence the pattern participants go on to use[28]. It has been argued that generalisation research needs to take into account higher-level cognitive processes such as category-based induction, inferential reasoning and representation of conceptual knowledge[21]. Function learning literature suggests that linear functions are more easily available to participants[50], providing one possible explanation for monotonic generalisation patterns that linearly map shape spikiness to outcomes. However, newer work shows that humans can flexibly learn complex functions[59]. Given our simple stimulus space, complexity likely did not drive generalisation patterns, and further research is needed to understand what drives the choice of a specific pattern. Relatedly, we note that our learning task did not include a CS-, a stimulus never paired with an aversive outcome. Including a CS- can induce differential conditioning and has been shown to impact the shape of generalisation gradients, for instance leading to so-called peak-shifts[68].

## Conclusion
In summary, our results suggest variability in the extent to which individuals rely on value-related processes to generalise aversive outcome expectancies to similar stimuli. Further, we show that anxiety is related to increased generalisation to more psychologically distant, i.e. more ambiguous, stimuli and that this increased generalisation is related to the tendency to rely on value-based generalisation rather than impoverished perceptual discriminability alone.

## Data availability
The behavioural data generated in this study have been deposited to GitHub and are openly accessible here: https://github.com/luiantav/vp_gen_cp. The raw behavioural data have been anonymized and are stored in a publicly available repository[69]: https://github.com/luiantav/vp_gen_rawdata. The code for the task used to collect the data has been deposited in a separate GitHub repository: https://github.com/luiantav/vp_gen_task.

## Code availability
The code used to derive statistical results is stored in the same GitHub repository as the data (https://github.com/luiantav/vp_gen_cp) and archived on Zenodo[70]. This repository further includes instructions to reproduce the results, including a dedicated computational virtual environment in R.

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

## Acknowledgements

Funding: LV was funded by the International Max Planck Research School on the Life Course (LIFE, www.imprs-life.mpg.de; participating institutions: Max Planck Institute for Human Development, Freie Universität Berlin, Humboldt-Universität zu Berlin, University of Michigan, University of Virginia, University of Zurich). BS was supported by DFG grant 462752742 and ERC grant 101000972. NWS was funded by an Independent Max Planck Research Group grant awarded by the Max Planck Society (M.TN.A.-BILD0004), the Federal Ministry of Education and Research (BMBF) and the Free and Hanseatic City of Hamburg under the Excellence Strategy of the Federal Government and the Länder and a Starting Grant from the European Union (ERC-StG-REPLAY-852669). OZ was supported by a Max Planck Research Group grant awarded by the Max Planck Society (M.TN.A.-BILD0004) to NWS and ERC Preparatory Fellowship awarded to O.Z. by Bielefeld University. The funders had no role in study design, data collection and analysis, decision to publish or preparation of the manuscript.

## Author contributions

The following list of author contributions is based on the CRediT taxonomy. Conceptualisation: L.V., O.Z, N.W.S; Data curation: L.V.; Formal analysis: L.V., O.Z., N.W.S.; Funding acquisition: L.V, N.W.S.; Investigation: L.V.; Methodology: L.V, O.Z., N.W.S.; Project administration: L.V., O.Z.; Software: L.V., O.Z. N.W.S.; Resources: N.W.S; Supervision: O.Z., N.W.S., B.S.; Validation: L.V., O.Z.; Visualisation: L.V.; Writing - original draft: L.V., O.Z., N.W.S.; Writing- review & editing: L.V., O.Z, N.W.S., B.S.

## Funding

## Competing interests

The authors declare no competing interests.
