## [Transparent Peer Review file · Communications Psychology]

Increased generalisation in trait anxiety is driven by aversive value transfer

Corresponding Author: Dr Ondrej Zika

Version 0:

Decision Letter:

Dear Dr Zika,

Thank you for your patience during the peer-review process. Your manuscript titled "Increased generalisation in trait anxiety is driven by aversive value transfer not reduced perceptual discrimination" has now been seen by 1 reviewers, and I include their comments at the end of this message. They find your work of interest but raised some important points. We are interested in the possibility of publishing your study in Communications Psychology, but would like to consider your responses to these concerns and assess a revised manuscript before we make a final decision on publication.

We therefore invite you to revise and resubmit your manuscript, along with a point-by-point response to the reviewers. Please highlight all changes in the manuscript text file.

We contacted Reviewer #3 from the previous round of evaluations and solicited feedback from another one of the former reviewers how they view the remaining points of expert disagreement. Asking reviewers for their opinion on each others' reports is a step we regularly take at the journal when we feel that the editorial decision will benefit from additional expert input, especially when reviewer reports suggest strongly different outcomes. While the reviewer who provided additional feedback agreed with some of Reviewer #3's concerns, they take the view that these do not invalidate the computational approach. However, they also expressed strong support for a renaming of the model and highlighted the necessity to give due consideration to alternative accounts.

On the basis of this combined feedback, we ask that you revise the framing of the computational model. Credible alternative accounts of underlying psychological processes need to be discussed in the Limitations. Due credit should be given to strictly relevant literature.

As you revise the manuscript in response to these issues, please also implement all requests in the attached Mandatory Revision Requests document. All requirements listed in this document need to be fully met, or the work will be returned to you for further revisions without peer review. This workflow is in place to increase the likelihood that the paper will be accepted for publication. It reduces the number of rounds of revision (and review) and ensures that the reviewers vet a version of the article that is compliant with journal policies. If you have any questions regarding the required revisions, please contact the journal prior to resubmission to avoid a negative outcome.

Please submit the following items:

- Revised manuscript
- Point-by-point response to the referees' comments
- Mandatory Revision Requests Table (attached).

- Cover letter (as a separate document)

via this link: Link Redacted .

** This url links to your confidential home page and associated information about manuscripts you may have submitted or are reviewing for us. If you wish to forward this email to co-authors, please delete the link to your homepage first **

Best regards,

Troy Lui

Troy Lui, PhD
Associate Editor
Communications Psychology

REVIEWER REPORTS:

Reviewer #3 (Remarks to the Author):

I previously reviewed an earlier version of this manuscript when it was submitted to Nature Communications, and I appreciate the authors' efforts in addressing several of the concerns raised during that process (although some key issues remain unaddressed). The paper's main strength lies in demonstrating that generalization gradients may arise from three distinct response profiles rather than two. This is a novel and interesting finding, and it naturally raises questions about the underlying mechanisms.

However, the central issue remains the use of mechanistic interpretations to explain what are, in essence, descriptive response patterns. This creates a risk of implying causal processes that the current data cannot support. Specifically, the interpretation of the binary responding pattern as reflecting a purely "perceptual model" is problematic. An alternative — and equally plausible — explanation is that participants rely on the same similarity-based function as in the Gaussian model but apply a binary decision rule on top of it (while still being aware that the stimuli differ). This would yield the same behavioral pattern without invoking perceptual confusion mechanisms. Thus, labeling this pattern as a "perceptual model" rests on a strong mechanistic assumption that the present data do not allow one to test or validate.

I therefore recommend reframing the manuscript to focus on identifying and characterizing distinct descriptive generalization profiles, without making mechanistic claims. Concretely, renaming the "perceptual model" to "binary model" would align its label with the descriptive nature of the other two models ("monotonic" and "Gaussian"), and mechanistic interpretations could be reserved for a more cautious, explicitly speculative discussion section. As currently written, referring to one model as "perceptual" implies the presence of specific psychological processes that are neither directly tested nor uniquely supported by the data.

Below I outline several (minor) remaining or newly arising concerns.

1. The hybrid model remains insufficiently described, as no full equations are provided. Based on the current text, my understanding is that for a given stimulus (e.g., a GS), there is a certain probability of misperceiving it as the CS+. If so, the response equals the expected CS+ value from the end of conditioning. If not, the expected response is computed via a generalization function scaled by α and the CS+ value. However, confusion arises when the stimulus is the CS+ itself. A CS+ has a probability of being correctly identified as the CS+, which produces the expected CS+ value. But if the CS+ is "misidentified," the generalization function yields $G = 1$ because the distance between the stimulus and itself is zero. Thus, the model predicts a full CS+ response in both cases. This is theoretically puzzling, as it implies identical predictions regardless of whether the CS+ is perceived correctly or incorrectly, whereas Fig. 4b shows substantial variability in CS+ responses. To resolve this, all hybrid model equations should be reported in full. As it stands, the implementation is difficult to reconstruct, and a complete formal specification is needed to evaluate its structure, assumptions, and internal coherence.

2. Inconsistent similarity functions across models - The authors argue that Gaussian-shaped gradients could emerge either from perceptual confusion with binary responding or from similarity-based value processes. Yet the models implement

similarity in fundamentally different ways: the so-called perceptual model uses a power function, whereas the value-based Gaussian model uses a Gaussian function. This discrepancy undermines model comparison, as fit differences may reflect the choice of similarity function rather than differences in psychological processes. A fair comparison requires using the same similarity function across models so that interpretative differences depend on model structure rather than on implementation choices.

3. Figure 3A is difficult for readers to understand in its current form. A clearer explanation in the caption or main text would greatly improve interpretability, especially for readers less familiar with this modeling framework.

4. The key analysis linking trait anxiety to the generalization parameter λ appears only within a complex three-way interaction — while the direct, theoretically most important correlation is not significant. The authors justify the interaction by stating that all parameters shown to influence λ were included. However, following this logic, it is unclear why reinforcement rate is included whereas stimulus discriminability — shown by the authors to affect λ — is omitted. This inconsistency raises concerns about the validity of these analyses. Moreover, higher-order interactions are notoriously difficult to interpret, replicate, and justify theoretically. The present interaction is not unpacked (e.g., how trait anxiety relates to λ across reinforcement conditions), leaving its robustness and meaning unclear. Given that the conclusions depend on an indirect effect emerging only through this complex interaction, and given uncertainty about why particular predictors were included or excluded (or if included would comprise a 4-way interaction), I recommend omitting these exploratory analyses and focusing on the direct correlations.

5. The relatively low prevalence of the “perceptual”/binary model may not be surprising. Unlike many generalization studies, the authors titrated stimulus discriminability to individual perceptual thresholds. This likely reduced perceptual confusion in their task, potentially explaining why binary-like response profiles occur less frequently than in earlier work. This interpretation would be worth noting explicitly, and future research should explore this possibility further.

6. The discussion of monotonic gradients in the introduction is overly simplistic. Recent work (e.g., [3,4]) shows that shifts in the memory representation of the CS+ can yield monotonic-like memory gradients that closely align with US expectancy patterns, providing a perceptual confusion account for these patterns. The authors’ framing does not acknowledge that such alternative accounts exist.

7. By forcing the CS+ representation in their models to be veridical, the authors overlook substantial evidence from fear generalization and perceptual categorization research showing that CS memories are imprecise, vary across individuals, and can shift relative to the actual stimulus [1,2,5]. Allowing for such variation would bring the assumed mechanisms more in line with current theory. This issue also relates directly back to the overarching concern: the authors cannot disentangle perceptual and value-based mechanisms with the present data, yet the manuscript currently attributes monotonic gradients exclusively to a value-based process. Given recent evidence that similar patterns can emerge from perceptual processes, mechanistic inferences should be avoided.

I understand that this would require substantial rewriting of the manuscript, but in my view, these changes are necessary for the paper to reach its full potential. In such a revised form, the study would represent a valuable, interesting, and more nuanced contribution to the literature.

References

- [1] Zaman J, Struyf D, Ceulemans E, Beckers T, Vervliet B. Probing perceptual mechanisms of fear generalization. *Sci Rep* n.d.
- [2] Zaman J, Struyf D, Ceulemans E, Vervliet B, Beckers T. Perceptual errors are related to shifts in generalization of conditioned responding. *Psychol Res* 2020;1–26. doi:10.1007/s00426-020-01345-w.
- [3] Zaman J, Yu K, Lee JC. Individual Differences in Stimulus Identification, Rule Induction, and Generalization of Learning. *J Exp Psychol Learn Mem Cogn* 2023. doi:10.1037/xlm0001153.
- [4] Zaman J, Yu K, Verheyen S. The idiosyncratic nature of how individuals perceive, represent, and remember their surroundings and its impact on learning-based generalization. *J Exp Psychol Gen* 2023;152:2345–2358. doi:10.1037/xge0001403.
- [5] Zenses A, Lee JC, Plaisance V, Zaman J. Differences in perceptual memory determine generalization patterns. *Behav Res Ther* 2021;136:103777. doi:10.1016/j.brat.2020.103777.

Version 1:

Decision Letter:

Dear Dr Zika,

Your manuscript titled "Increased generalisation in trait anxiety is driven by aversive value transfer" has now been assessed editorially. In light of their advice I am delighted to say that we are happy, in principle, to publish a suitably revised version in Communications Psychology.

We therefore invite you to revise your paper one last time to address the remaining concerns of our reviewers and a list of editorial requests. At the same time we ask that you edit your manuscript to comply with our format requirements and to maximise the accessibility and therefore the impact of your work.

EDITORIAL REQUESTS:

SUBMISSION INFORMATION:

OPEN ACCESS:

*** TRANSPARENT PEER REVIEW:** Communications Psychology uses a transparent peer review system. On author request, confidential information and data can be removed from the published reviewer reports and rebuttal letters prior to publication. If you are concerned about the release of confidential data, please let us know specifically what information you would like to have removed. Please note that we cannot incorporate redactions for any other reasons.

*** CODE AVAILABILITY:** All Communications Psychology manuscripts must include a section titled "Code Availability" at the end of the methods section. We require that the custom analysis code supporting your conclusions is made available in a publicly accessible repository at this stage; please choose a repository that generates a digital object identifier (DOI) for the code; the link to the repository and the DOI must be included in the Code Availability statement. Publication as Supplementary Information will not suffice.

*** DATA AVAILABILITY:**

All Communications Psychology manuscripts must include a section titled "Data Availability" at the end of the Methods

section. More information on this policy, is available in the Editorial Requests Table and at <http://www.nature.com/authors/policies/data/data-availability-statements-data-citations.pdf>.

Link Redacted

Best regards,

Troby Lui

Troby Lui, PhD
Associate Editor
Communications Psychology

Title: Increased generalisation in trait anxiety is driven by aversive value transfer

Authors: Lianta Verra, Bernhard Spitzer, Nicolas W. Schuck and Ondrej Zika

Point-by-point response

Reviewer #3 (Remarks to the Author):

I previously reviewed an earlier version of this manuscript when it was submitted to Nature Communications, and I appreciate the authors' efforts in addressing several of the concerns raised during that process (although some key issues remain unaddressed). The paper's main strength lies in demonstrating that generalization gradients may arise from three distinct response profiles rather than two. This is a novel and interesting finding, and it naturally raises questions about the underlying mechanisms.

We thank the reviewer for the time spent on the manuscript. We have been informed that due to a miscommunication the reviewer has not seen our responses to the points they have raised in the previous round of revisions. We sincerely regret this, as we are aware that this must have raised questions and additional work on both our and the reviewer's side. Further, we apologise if the reviewer got the impression that we did not address their concerns, and remain committed to address all open points. Below, we reply to all suggestions. For completeness, we also attach our responses to the second round of revisions in Nature Comms as an additional document

"Verra_et_al_Previous_Revision_Nature_Comms_Round2.pdf".

R3-1. However, the central issue remains the use of mechanistic interpretations to explain what are, in essence, descriptive response patterns. This creates a risk of implying causal processes that the current data cannot support. Specifically, the interpretation of the binary responding pattern as reflecting a purely "perceptual model" is problematic. An alternative — and equally plausible — explanation is that participants rely on the same similarity-based function as in the Gaussian model but apply a binary decision rule on top of it (while still being aware that the stimuli differ). This would yield the same behavioral pattern without invoking perceptual confusion mechanisms. Thus, labeling this pattern as a "perceptual model" rests on a strong mechanistic assumption that the present data do not allow one to test or validate. I therefore recommend reframing the manuscript to focus on identifying and characterizing distinct descriptive generalization profiles, without making mechanistic claims. Concretely, renaming the "perceptual model" to "binary model" would align its label with the descriptive nature of the other two models ("monotonic" and "Gaussian"), and mechanistic interpretations could be reserved for a more cautious, explicitly speculative discussion section. As currently written, referring to one model as "perceptual" implies the presence of specific psychological processes that are neither directly tested nor uniquely supported by the data.

The reviewer makes a good point in raising the possibility that participants could be applying a binary response rule, after generalising value following a gradual value function. To test this possibility we fitted an alternative value model that additionally transforms beliefs stemming from value generalisation V_s before answering using a sigmoid transformation with fitted slope parameter k .

$$y_{transformed} = \frac{1}{1+e^{-k(V_s-0.5)}}$$

High values of k therefore give the value model the ability to respond in a binary manner. This model fits better than the perceptual or value-based model in only a small proportion of cases, 11.67%, as can be seen in Figure 1 below. When assessing if these gradients best fit by the alternative model were previously classified as value-based or perceptual we find that only 3.06% of these gradients were previously classified as perceptual. This shows that the sigmoid transformation improved the fit for those gradients that were already classified as value-based and that binary response distributions in our case seems to best be explained by a perceptual account.

Figure 1: % best fit by the perceptual model, value model and value model including a sigmoid transformation of values before response.

That being said, the binary response pattern is a plausible prediction from a perceptual point of view. The idea of the perceptual model producing binary responses is not new but has been brought forward by Struyf, Zaman et al. (2015) as one possible mechanism of generalisation (Please see paper figure 2B). The model is theory driven and provides a mechanistic explanation of how these responses result from misperception. To our knowledge we are the first to formally test this hypothesis and to show that in some participants this mechanism seems to explain generalisation gradients best.

Finally, we want to highlight that the instructions in our task promoted gradual responding i.e. “How likely do you think it is that this “flower” screams at you?” and not to discriminate the CS+ i.e. “Was the seen stimulus the CS+?” or make a response/avoidance response i.e. “Would you approach this flower?”.

We nevertheless agree with the reviewer that more attention should be given to alternative explanations of binary response patterns in our manuscript. We now discuss the possibility of alternative mechanisms resulting in binary responses on pages 19, 20 and 27, and, importantly, adapted our manuscript title. We have also added the binary choice model as an alternative model to the Supplement. All changes to the manuscript are reported below.

Title change:

Increased generalisation in trait anxiety is driven by aversive value transfer

Results

To capture differences in the underlying generalisation mechanisms (perceptual vs value-based), and in the generalisation pattern (monotonic vs Gaussian-like), we developed two computational models. Both models assume that participants' ratings responses reflect their underlying outcome expectancies. The perceptual model assumes that generalisation occurs as a consequence of impoverished discriminability. In this model, the perceived stimulus on a given trial is assumed to be a single sample from a probability distribution centered on the presented stimulus. The probability of perceiving a particular stimulus decreases with its distance from the presented one, scaled by an uncertainty parameter (ρ). Two predictions arise from this account: First, rating response distributions to generalisation stimuli should be bi-modal, reflecting that the same stimulus was sometimes perceived as different from the CS+, resulting in a rating of 0%, and on other occasions confused with the CS+, resulting in a rating that matches CS+ expectations (Fig. 2a). Under this hypothesis, stimuli close to the CS+ are expected to be misperceived as the CS+ more frequently which, on average, may give rise to a Gaussian-like pattern (Fig. 2c, *left*). We note that other factors can in principle result in a binary response pattern, such as participants employing a binary response strategy, an alternative explanation we tested in the Supplementary (see below). Second, perceptual mechanisms cannot account for monotonic generalisation patterns, since the confusability with the CS+ decreases with perceptual distance. Our model does not assume systematic shifts in perception or memory, such as misremembering and misidentifying the most extreme stimulus as the CS+. Hence, perceptual generalisers are expected to show Gaussian-like response patterns when averaged, but bi-modal distributions on a trial wise level.

[...]

We additionally tested a hybrid model combining value generalisation (value model) with misperception of stimuli (perceptual model) as illustrated in Fig. 2a. Model comparison favored the value-based (76.90%) over the hybrid (8.1%) and perceptual (15.0%) models, suggesting that the value component explains most variability in behaviour. We further tested a binary choice model to test if the binary response pattern observed in our data could reflect a binary decision rule. 11.67% of gradients were best fit by this model, out of which only 3.06% were previously best fit by the perceptual model (see Supplementary Materials: Alternative Models).

Discussion

[...]

While our modeling approach focuses on dissociating behavioural signatures of perceptual and value-based generalisation, our models can be combined to account for joint value generalisation and misperception; an example is provided in the Supplement (section Hybrid model). Note that our perceptual and value models both assume that participants' value beliefs are directly expressed in their ratings. To test this assumption, we fitted an alternative model that includes a transformation of value estimates in a manner that can result in binary ratings. This model, however, fitted the data less well.

In addition, we acknowledge that other variables might have introduced noise to participants' responses, such as poor memory or attention. Recent work has shown that participants can systematically misremember which stimulus was associated with a scream^{9,30} and that such memory bias can account for differences in generalisation pattern. While we did not observe systematic shifts in CS+ memory, we cannot dissociate noise due to poor memory or attention effects from perception in the current task design.

Finally, we acknowledge that our experiment did not systematically manipulate factors that could impact the balance of perceptual and value based generalisation, and hence relies on describing pre-existing differences between participants of unknown origin. To strengthen mechanistic insights, future work should therefore seek to experimentally manipulate process-relevant variables. In this context we note that our participants were instructed to rate how likely a scream is, which could have led to a higher prevalence of value-based responding. Such a question, however, can best be solved by experimental manipulation.

Supplement

Binary choice model

[the text is identical to what we report above when the model is described]

Minor

R3-2: The hybrid model remains insufficiently described, as no full equations are provided. Based on the current text, my understanding is that for a given stimulus (e.g., a GS), there is a certain probability of misperceiving it as the CS+. If so, the response equals the expected CS+ value from the end of conditioning. If not, the expected response is computed via a generalization function scaled by α and the CS+ value. However, confusion arises when the stimulus is the CS+ itself. A CS+ has a probability of being correctly identified as the CS+, which produces the expected CS+ value. But if the CS+ is "misidentified," the generalization function yields $G = 1$ because the distance between the stimulus and itself is zero. Thus, the model predicts a full CS+ response in both cases. This is theoretically puzzling, as it implies identical predictions regardless of whether the CS+ is perceived correctly or incorrectly, whereas Fig. 4b shows substantial variability in CS+ responses. To resolve this, all hybrid model equations should be reported in full. As it stands, the implementation is difficult to reconstruct, and a complete formal specification is needed to evaluate its structure, assumptions, and internal coherence.

We would like to point the reviewer to the Supplementary Information, where the equations have been added in full as part of the changes to the previous round of revisions (this was part of the second revision in NC which did not reach the reviewer). Being a combination of the perceptual and value model, the equations are also equivalent to the equations of the individual models.

We also want to note that the description of the hybrid model above is partially incorrect. G depends on three factors, the distance of a perceived stimulus from the CS+, the generalisation strength λ and the pattern ω . Here we highlight that the distance represents the distance of a perceived stimulus from the CS+, irrespective of the stimulus shown. For example:

Stimulus shown	Stimulus perceived	Distance from CS+
GS1	GS1 (correctly perceived)	1
GS1	CS+ (misperceived)	0
CS+	CS+ (correctly perceived)	0
CS+	GS1 (misperceived)	1

It follows that if a CS+ is shown, but misperceived as a GS (let's assume GS1), then the distance of the misperceived stimulus GS1 and the CS+ is 1. This is the distance that is used to calculate G . $G=1$ only applies when the perceived stimulus is the CS+ (distance of perceived stimulus and CS+ = 0), irrespective of which stimulus was shown. We hope that this has helped to further clarify the hybrid model. We have further specified this in the Supplement and report the respective section below.

Supplementary Information:

Hybrid model of generalisation

Specifically, we add the probabilistic component of the perceptual model to the value generalisation model such that when participants misperceive a stimulus, the generalised value for the misperceived stimulus q is reported rather than the value for the presented stimulus s .

Similar to the perceptual model, the probability $P(q)$ to perceive a stimulus q on a given trial depends on the perceptual discriminability of neighboring stimuli ρ and the distance of q from the presented stimulus s . $P(q)$ decreases with increasing distance of a stimulus q from the presented stimulus s , following a multinomial distribution over stimuli given by

$$\begin{aligned}
 P(q|s) &= \rho^{d_{|s-q|}} && \text{if } p \neq q \\
 P(q|s) &= 1 - \rho && \text{if } p = q
 \end{aligned}$$

where $d_{|s-q|}$ corresponds to the distance between q and s .

Similar to the value model, the value for a specific stimulus V depends on a participants' generalisation function G and the generalisation tendency λ : $V = V_{CS+} * G_{\Omega}$. Analogous to the value model, the value function is specified as follows:

$$\text{If } \Omega = 1 : G_{\text{gaussian}} = \frac{2}{1 + e^{\frac{d_s^2}{\lambda}}}$$

$$\text{If } \Omega = 0 : \begin{cases} G_{s_{\{1:4\}}, \text{linear}} = 1 + \left| 1 - G_{s_{\{1:4\}}, \text{linear}} \right| \\ G_{s_{\{5:9\}}, \text{linear}} = G_{s_{\{5:9\}}, \text{gaussian}} \end{cases}$$

d represents the distance of the perceived stimulus from the CS+. Analogous to the value model, the general tendency to over- or under-estimate value is specifying with a constant relative offset parameter $\alpha \in [-1, 1]$. Critically, the trial-wise predictions of the hybrid model correspond to the generalised values of the perceived stimulus q , not as in the value model the value of the shown stimulus s : $y = V_q + \alpha$

R3-3. Inconsistent similarity functions across models - The authors argue that Gaussian-shaped gradients could emerge either from perceptual confusion with binary responding or from similarity-based value processes. Yet the models implement similarity in fundamentally different ways: the so-called perceptual model uses a power function, whereas the value-based Gaussian model uses a Gaussian function. This discrepancy undermines model comparison, as fit differences may reflect the choice of similarity function rather than differences in psychological processes. A fair comparison requires using the same similarity function across models so that interpretative differences depend on model structure rather than on implementation choices.

Thank you for your question. The question has been raised in a previous review and we regret that the Reviewer has never seen our response, which unfortunately did not get passed on during the manuscript transfer process due to a file mislabeling error. Below please find our reply, largely copied from the last revision.

The specification of similarity as a value function (Monotonic, Gaussian) vs. probabilistic misperception reflects two different processes, namely a function learning process (value model) and a probabilistic perceptual process (perceptual model). As such, the two functions are intended to be different by design. We are sorry that this misunderstanding arose. Both functions operate on similarity (same across functions) but the mechanism is different. While the generalisation function in the value model operates on the value level and reflects value propagation to similar stimuli, the probability of misperception function in the perceptual model represents a probability distribution modeling which stimulus is perceived. The latter considers perceived similarity of each participant following a process of stimulus titration. We have illustrated this point in Figure 2a (previously 3a), where for each model we show the underlying value assumption as well as the probability of stimulus misperception. The

key aspect is that while a person would hold the value function as a cognitive variable, the perceptual distribution comes from interaction between the observer and the environment.

R3-4. Figure 3A is difficult for readers to understand in its current form. A clearer explanation in the caption or main text would greatly improve interpretability, especially for readers less familiar with this modeling framework.

Thank you for your useful comment. To improve clarity we have made changes to the caption of Figure 2A (previously Figure 3a) and to the figure itself to clarify the main components of our illustration:

Figure 2: Model predictions. a Visualization of value-based and perceptual mechanisms across the three models: perceptual, value-based and hybrid models. The top row illustrates the value functions across the three models, which for the perceptual model (top left) is a stick function centered on the CS+ (value directly acquired through learning). The value and hybrid models, in contrast, have Gaussian or linear value functions that assign value to stimuli which were not paired with the aversive outcome during learning (value function parameterized, see panel b). The bottom row shows the probability with which stimuli are misperceived. The perceptual model assumes a probabilistic confusion of each stimulus with neighboring stimuli, which we empirically derived from the titration task. Shown are the example probabilities of all test stimuli given GS1 (flower on x-axis) was shown. The value model does not assume any perceptual confusion, and therefore assigns 100% probability to the observed stimulus. The hybrid model has the same misperception probability function as the perceptual model. The ratings participants provide are assumed to reflect the value of the perceived stimulus for each model (but see Methods). Predicted response distributions are shown in panel c. b Effects of parameters in the value function G of the value and hybrid models. Each model has a width parameter λ which increases generalisation

and a shape parameter Ω which determines whether the function is linear ($\Omega = 0$) or Gaussian-like ($\Omega = 1$). c Predicted response distributions of the perceptual and value models (the latter separately for Gaussian-like or monotonic value functions). The stimulus average for both perceptual generalisation and Gaussian-like value generalisation follows a Gaussian-like shape. d Model recovery based on BIC (Bayesian Information Criterion) for the perceptual model (P) and value model (V).

R3-5. The key analysis linking trait anxiety to the generalization parameter λ appears only within a complex three-way interaction — while the direct, theoretically most important correlation is not significant. The authors justify the interaction by stating that all parameters shown to influence λ were included. However, following this logic, it is unclear why reinforcement rate is included whereas stimulus discriminability — shown by the authors to affect λ — is omitted. This inconsistency raises concerns about the validity of these analyses. Moreover, higher-order interactions are notoriously difficult to interpret, replicate, and justify theoretically. The present interaction is not unpacked (e.g., how trait anxiety relates to λ across reinforcement conditions), leaving its robustness and meaning unclear. Given that the conclusions depend on an indirect effect emerging only through this complex interaction, and given uncertainty about why particular predictors were included or excluded (or if included would comprise a 4-way interaction), I recommend omitting these exploratory analyses and focusing on the direct correlations.

We thank the reviewer for their comment and we agree that a three-way interaction between lambda, reinforcement rate and offset is complex and difficult to unpack. We would however like to note that such an interaction is expected. Critically, the width of the generalisation function depends on the height and both reinforcement rate and offset impact the height.

To better explain the 3-way interaction, we included Figure 5d (replicated below) which illustrates this complex point. Specifically, the figure shows how lambda differs under different reinforcement conditions and offsets and how this is impacted by anxiety. To provide a general intuition for the role of anxiety in generalization, we also present empirical means for median-split anxiety - higher anxiety tends to have higher lambda.

While the reviewers are correct in noting that discriminability impacts generalisation (lambda), our analyses indicate that it does not affect the offset. For this reason, we have initially not included it in our model. To address the reviewers specific concern, we ran an alternative model including both reinforcement rate and discriminability and found that including discriminability did not change the results. Specifically, we ran the following model:

*$\text{Lambda} \sim \text{trait_anxiety} * \text{offset} * \text{reinforcement_rate} + \text{trait_anxiety} * \text{offset} * \text{discriminability}$*

As the reviewer correctly states, this model shows an expected effect of discriminability on lambda, $\chi^2(2) = 35.31$, $p < 0.001$, $\eta_p^2 = 0.2$, CI = [0.1, 0.31], but no statistically significant interaction with trait anxiety. We have now updated our results to include this model for completeness and report changes below.

Results

We next assessed the association between the model derived generalisation parameter λ and trait anxiety and found no statistically significant main effect of trait anxiety on generalisation width, $\beta = 0.004$, $\chi^2(1)=0.00$, $p = 0.98$, $\eta_p^2 = 0.00$, $CI = [0.00,0.00]$. As reinforcement rate, discriminability and offset were found to impact λ , we specified a model of λ including these variables and their interaction with trait anxiety. This model revealed a significant 3-way interaction of trait anxiety, α and reinforcement rate, $\chi^2(2) = 14.12$, $p < 0.001$, $\eta_p^2 = 0.02$, $CI = [0.00,0.05]$, suggesting that trait anxiety impacts λ differently depending on α and the reinforcement condition. Figure 5d illustrates this relationship, showing an effect of TA on generalisation strength after accounting for other variables that impact λ .

Figure 5: **Trait anxiety effects on generalisation.** a Transformed expectancy ratings by trait anxiety (TA) split by median into high vs. low trait anxious participants. Lines show mean ratings per anxiety group ($n=70$ per group), shaded areas show standard errors of the mean. The figure insert shows the effect of trait anxiety on generalisation ratings which increases with increasing distance of a stimulus from the CS+, $\beta = 0.03$, $\chi^2(1)=25.25$, $p < 0.001$. b Histogram of STICSA scores ($n=140$ participants). c Relationship of trait anxiety and relative model fit where trait anxiety was associated with better relative fit of the value model across generalisation patterns, $\beta = 3.37$, $\chi^2(1)=5.24$, $p = 0.022$, $\eta_p^2 = 0.006$ $[0.0, 0.05]$. Each dot represents one participant ($n=140$), the relative model fit was however based on fits per task condition ($n=840$). We show the correlation of trait anxiety and relative model fit with the 95% confidence intervals obtained through bootstrapping. The dashed line represents the lowest

possible STICSA value. d Association of model parameters λ , α and the experimental reinforcement rate manipulation and their relation to anxiety. Each dot represents one participant ($n=140$), the lines represent the association of α with λ by the anxiety group (median split into high and low). The figure inserts to the left of the scatterplots show the mean lambda by anxiety group. Bars represent the standard error of the mean.

R3-6. The relatively low prevalence of the “perceptual”/binary model may not be surprising. Unlike many generalization studies, the authors titrated stimulus discriminability to individual perceptual thresholds. This likely reduced perceptual confusion in their task, potentially explaining why binary-like response profiles occur less frequently than in earlier work. This interpretation would be worth noting explicitly, and future research should explore this possibility further.

The reviewer makes an interesting point. Studies without titration of the perceptual spaces will likely lead to higher **variability** in perceptual effects, but whether they would result in **overall** reduced perceptual confusion is unclear. We agree that this would have to be systematically tested. In either case, we would argue that for any inquiry into perceptual processes it is essential to understand the baseline confusability between stimuli beforehand, as done in our study.

Titration allows us to characterize the rate of misperception, and ensures that participants do not get better at perceiving stimuli. In comparison with other studies, more perceptual confusion would occur in studies where similarity between stimuli is higher. However, our work does not aim to demonstrate more or less generalisation compared to different tasks, which, as the reviewer correctly notes, differ on a range of task parameters. We show that even with constant discriminability, participants differ in the extent to which they generalise value.

For clarity, we also note that in our task discriminability was titrated to fixed target levels, not to participants’ perceptual thresholds. This is indeed unlike many generalisation studies and allows us to hold discriminability constant across participants, even if perceptual thresholds might differ. We would additionally want to emphasise that we do not think of ~15% best fit by a pure perceptual model as low. In contrast, we find it interesting and novel that pure perceptual generalisation seems to explain generalisation best in some participants. We have added discussion of these differences to the paper.

Introduction

In scenarios where stimuli vary along a perceptual dimension (e.g., to what degree a new dog resembles the dog that bit us), participants may sometimes perceive a novel stimulus as identical to the previously presented CS+, and treat it as such. Prior work has shown that misidentification increases generalisation responses, depends on the similarity of a stimulus to the CS+, and varies substantially across individuals.^{5,6,8,9} These findings highlight the importance of accounting for perception in generalisation.

Discussion

Stimuli in many generalisation experiments are based on physical stimulus features rather than on subjects' perception^{4,7} which often makes it difficult to control for discriminability. Recent modeling work accounting for stimulus perception has shown how similar generalisation tendencies can produce different responses depending on how stimuli were perceived¹⁷. Here we manipulate stimulus discriminability and show how discriminability influences generalisation gradients, adding to findings associating an individuals' perceptual acuity to the degree of generalisation. Because prior studies have used stimulus spaces with differing overall stimulus similarity, comparisons across studies are difficult; we would however expect that spaces titrated to lower discriminability levels would result in overall higher misperception.

R3-7. The discussion of monotonic gradients in the introduction is overly simplistic. Recent work (e.g., [3,4]) shows that shifts in the memory representation of the CS+ can yield monotonic-like memory gradients that closely align with US expectancy patterns, providing a perceptual confusion account for these patterns. The authors' framing does not acknowledge that such alternative accounts exist.

As with the previous points, we have addressed this item in the previous revision. To provide a more nuanced and balanced account, we introduce the possibility of misidentification in the context of monotonic gradients on page 5 of the introduction as well as in the discussion on page 28 and 32. Specifically we cite the two works that the reviewer is suggesting. We report the corresponding paragraphs below. Further, we would like to briefly summarise why in our task systematic memory shifts and misidentification in the context of monotonic gradients are unlikely.

- A prediction of memory shifts implies that the value of the CS+ would remain similar, even if reported for a different stimulus (i.e. we speak of memory for stimulus identity, not value). This is contradicted by the fact that in our data the expectancy rating reported for the extreme stimulus is much **higher** than the CS+ value. It is therefore more likely that participants use a monotonic rule to extrapolate stimulus. Similar phenomena have been described in the function learning literature (Wu, Meder & Schultz, 2025). This is further supported by the feedback participants provided after completing the task. Specifically we asked participants: "*Did you worry that potentially there would be an **even more dangerous space flower than the super space flower that you have seen screaming?***". 76.47% of the participants with the majority of gradients classified as monotonic agreed to this statement compared to 20.63% in the gaussian group.
- Given that we know discriminability in our stimulus space, systematic misperception of the most extreme stimulus (GS4) as the CS+ is extremely unlikely. The probabilities are 0.0016 and 0.0256, for the two discriminability conditions respectively (80% and 60%). It is therefore unlikely that about half of the participants would have systematically misperceived/misremembered the most distant generalisation stimulus as the CS+.
- In our task, participants complete 6 rounds of titration, learning and generalisation (conditions). Even though we control for discriminability across conditions, we find variability of pattern within participants in about one quarter of our sample. This

indicates that despite the fact that perceptual discriminability was held constant, there can be differences in generalisation pattern.

- The works mentioned by the reviewer differ in a range of aspects from this paper. First, our design does not include a CS- (i.e. differential conditioning). Introducing a CS- can lead to monotonic patterns: participants are directly introduced to the idea of higher and lower conditioning strength, promoting extrapolation-like generalization responses, we discuss this on page 32. To hint at this possibility, in the cited work, stronger monotonic patterns seem to emerge in the differential case. Second, in the simple conditioning datasets, gradients labeled as linear are often flat, rather than monotonically increasing/decreasing. These could reflect a range of factors, from inattention to flat Gaussian-like value generalization, since the responses are not systematically higher on one end of the stimulus space. In this context, misidentification is a plausible interpretation of the data: if one cannot distinguish stimuli one cannot provide differential generalisation responses. However, we note that this is not the case in our data set where monotonic gradients due to misperception remain highly unlikely and the most plausible explanation for monotonic gradients is rule-based inference (i.e. “spikier/rounder flowers will be more threatening”).

Charley M. Wu, Björn Meder, Eric Schulz. 2025. Unifying Principles of Generalization: Past, Present, and Future. *Annual Review Psychology*. 76:275-302.
<https://doi.org/10.1146/annurev-psych-021524-110810>

Introduction

Monotonic gradients have been mainly reported in human participants^{26,27} and are thought to reflect abstract strategies that participants apply to generalise value. Such strategies can be thought of as a form of function learning where stimuli are associated with outcomes based on some inferred function. These generalisation functions can, for instance, be monotonic, reflecting a linear relationship, or Gaussian-shaped, when participants assume that outcomes scale with perceptual similarity. Alternative accounts have associated different generalisation patterns with differences in perceptual discriminability^{9,30}.

Discussion

We show that discriminability relates to the amount of generalisation, and that despite holding discriminability constant, one quarter of participants showed different generalisation patterns across task conditions. This suggests that individual differences in generalisation vary beyond what can be explained by perceptual differences alone. It further contrasts recent work suggesting perceptual differences as a pathway to understanding individual differences in generalisation patterns⁹ such as work relating more accurate CS+ identification with a tendency for similarity based generalisation³⁰.

[...]

Relatedly, we note that our learning task did not include a CS-, a stimulus never paired with an aversive outcome. Including a CS- can induce differential conditioning and has been

shown to impact the shape of generalisation gradients, for instance leading to so-called peak-shifts⁶⁹.

R3-8. By forcing the CS+ representation in their models to be veridical, the authors overlook substantial evidence from fear generalization and perceptual categorization research showing that CS memories are imprecise, vary across individuals, and can shift relative to the actual stimulus [1,2,5]. Allowing for such variation would bring the assumed mechanisms more in line with current theory. This issue also relates directly back to the overarching concern: the authors cannot disentangle perceptual and value-based mechanisms with the present data, yet the manuscript currently attributes monotonic gradients exclusively to a value-based process. Given recent evidence that similar patterns can emerge from perceptual processes, mechanistic inferences should be avoided.

We agree with the reviewer in that memories are imprecise and plastic and that this can impact generalisation responses. We are however unsure if this point refers to effects of poor perceptual discriminability (“I am not sure if what I’m seeing is the CS+”) or poor memory (“I am unsure which stimulus was the CS+”) on generalisation, as these terms seem to be used interchangeably. While these are theoretically distinct, the titration performed in this task accounts for both processes. Discriminability in our task is known and controlled for, this allows us to make predictions and verify them in our data. On the other hand, memory and misperception are not dissociated by design. We have discussed this in the first round of revisions where we have specified that here we focus on misidentification giving rise to perceptual uncertainty which can be due to memory or perception. We have specified this distinction in our discussion on page 31. Regarding systematic memory shifts and differences in our work from the work cited by the reviewer we would like to refer to our answer to the previous point (R3-7).

Other changes as part of the mandatory revision requests:

- Changes to last sentence in Introduction:

In this work, we characterise value-based and perceptual contributions to generalisation within a controlled stimulus space and assess their contribution to anxiety-related differences in generalisation. We hypothesize that individuals vary in the extent and the pattern of value generalisation, and that higher trait anxiety is associated with stronger value generalisation.

- Changes to Abstract:

Anxiety has been linked to increased generalisation of threat expectations to perceptually similar stimuli. Such generalisation can arise either from a failure to distinguish threatening from non-threatening stimuli (perceptual mechanism) or from the transfer of learned values between stimuli (value-based mechanism). Yet, how

these mechanisms contribute to generalisation remains unclear. Here we assess how participants ($n = 140$) generalise outcome expectancies to perceptually similar stimuli, using personalised stimulus spaces. Computational modelling revealed that individuals differ in the extent to which they generalise value and in the underlying value function. We further found that stronger generalisation in trait anxiety was best explained by greater reliance on value transfer. In this work we characterise individual differences in generalisation of aversive stimuli and link stronger generalisation in trait anxiety to preferential reliance on value transfer.

- Added page numbers to manuscript
- Changed Figure order (to reflect new order of text i.e. methods precede results)
- Moved Figures and Figure legends to bottom of manuscript